# Development of a real-time on-road emission (ROE v1.0) model for street-scale air quality modeling based on dynamic traffic big data

Luolin Wu[1], Ming Chang[2], Xuemei Wang[2], Jian Hang[1], Jinpu Zhang[3], Liqing Wu[1]

[1]School of Atmospheric Sciences, Sun Yat-sen University, Guangzhou 510275, P. R. China

[2]Institute for Environmental and Climate Research, Jinan University, Guangzhou 510632, P. R. China

[3]Guangzhou Environmental Monitoring Center, Guangzhou 510030, P. R. China

*Correspondence to*: Xuemei Wang (eciwxm@jnu.edu.cn)

**Abstract.** Rapid urbanization in China has led to heavy traffic flows in street networks within cities, especially in eastern China, the economically developed region. This has increased the risk of exposure to vehicle-related pollutants. To evaluate the impact of vehicle emissions and provide an on-road emission inventory with higher spatial–temporal resolution for street-network air quality models, in this study, we developed the Real-time On-road Emission (ROE v1.0) model to calculate street-scale on-road hot emissions by using real-time big data for traffic provided by the Gaode map navigation application. This Python-based model obtains street-scale traffic data from the map application programming interface (API), which are open-access and updated every minute for each road segment. The results of application of the model to Guangzhou, one of the three major cities in China, showed on-road vehicle emissions of carbon monoxide (CO), nitrogen oxide ($NO_x$), hydrocarbons (HC), $PM_{2.5}$, and $PM_{10}$ to be $35.22 \times 10^4$ Mg/yr, $12.05 \times 10^4$ Mg/yr, $4.10 \times 10^4$ Mg/yr, $0.49 \times 10^4$ Mg/yr, and $0.55 \times 10^4$ Mg/yr, respectively. The spatial distribution reveals that the emission hotspots are located in some highway-intensive areas and suburban town centers. Emission contribution shows that the dominant contributors are light-duty vehicles (LDVs) and heavy-duty vehicles (HDVs) in urban areas and LDVs and heavy-duty trucks (HDTs) in suburban areas, indicating that the traffic control policies regarding duty trucks in urban areas are effective. In this study, the Model of Urban Network of Intersecting Canyons and Highways (MUNICH) was applied to investigate the impact of traffic volume change on street-scale photochemistry in the urban areas by using the on-road emission results from the ROE model. The modeling results indicate that the daytime $NO_x$ concentrations on national holidays are 26.5% and 9.1% lower than those on normal weekdays and normal weekends, respectively. Conversely, the national holiday $O_3$ concentrations exceed normal weekday and normal weekend amounts by 13.9% and 10.6%, respectively, owing to changes in the ratio of emission of VOCs and $NO_x$. Thus, not only the on-road emission, but other emissions should be controlled in order to improve the air quality in Guangzhou. More significantly, the newly developed ROE model may provide promising and effective methodologies for analyzing real-time street-level traffic emissions and high-resolution air quality assessment for more typical cities or urban districts.

## 1 Introduction

Rapid economic development and urbanization have led to the exponential growth in the number of vehicles in China has grown in recent years (National Bureau of Statistics of China, 2017). As one of the three major urban clusters, the Pearl River Delta (PRD) region, or its main city, Guangzhou, has experienced a significant increase in the number of vehicles. This increase has become the dominant contributor to carbon monoxide (CO), nitrogen oxide ($NO_x$), and hydrocarbon (HC) emissions (He et al., 2002; Zheng et al., 2009a), which in turn are causing more frequent and more severe public health problems in Chinese megacities (An et al., 2013). Previous studies have shown that on-road vehicle emissions can contribute approximately 22–52% of total CO, 37–47% of total $NO_x$, and 24–41% of total HC emissions detected in cities (Zhang et al., 2009; Zheng et al., 2009a, 2014; Li et al., 2017).

Reliable on-road emission inventories can be used as input data for the numerical air quality models which are applied to estimate the impact of on-road emissions on the urban air quality (Wang and Xie, 2009; He et al., 2016). For this purpose, the

realistic on-road vehicle emission inventory should be developed as the pollutant source. The two main methodologies used in recent years to establish such inventories are top-down and bottom-up techniques.

Top-down methods, such as that used in the MOBILE model devised by the US Environmental Protection Agency (EPA) and other similar macroscale models, first require information on vehicle population, vehicle kilometers travelled (VKT), and mean vehicle speed for an entire city to calculate the total amount of vehicular emissions. Then, the emissions are allocated to each grid cell utilizing parameters such as road density and road hierarchy (Saide et al., 2009; Jing et al., 2016; Liu et al., 2018). Many studies have adopted this method to develop city- or national-level vehicle emission inventories in China (Hao et al., 2000; Cai and Xie, 2007; Guo et al., 2007; Saide et al., 2009; Zheng et al., 2009a; Sun et al., 2016). However, the top-down inventories offer low-level spatial and temporal resolution because of the allocation method and input data used. Typically, the spatial allocation of a top-down inventory is based on the road network. The greater the road density and length, the higher the amount of emissions in the same grid. This allocation method simplifies the road emissions by assuming that every road of a specific road type (e.g., highway, arterial road, and local road) experiences the same traffic volume irrespective of its location. In addition, emission factors are considered to remain unchanged despite the traffic speed over the entire city, thereby leading to inaccurate results for the inventory. Moreover, some megacities (e.g., Guangzhou), have traffic control policies in place in certain specific urban areas, which implies that the emissions should differ across areas. Besides, the VKT data are usually provided on the yearly scale, which limits the temporal resolution of the inventory. For numerical modeling, the accuracy of the emission inventory may have a great impact on the simulation result because of the strong dependence of numerical models on it (Jing et al., 2016). This scale of the emission inventory may not reflect the real emission conditions for the on-road vehicles within the city, and thus, evaluations of traffic-related impacts on air pollution in complex situations such as street-level traffic flow are likely to be inaccurate (Huo et al., 2009).

Consequently, several studies have established higher-resolution inventories using the bottom-up approach. The main difference from top-down method is that bottom-up inventories are based on information from road segments. Therefore, spatial distribution is directly obtained from the input data, and spatial and temporal allocations are not required. Among the input data, the traffic data are crucial for establishing the inventories and determining their accuracy. Some previous studies have used traffic simulation models to obtain traffic speed or volume data of road networks (Pallavidino et al., 2014; Zhang et al., 2016; Chen et al., 2017; Ibarra-Espinosa et al., 2018). Based on the traffic model, the method could provide traffic data for each road from low-resolution average data. However, the results from such traffic models may not reflect reality, thereby reducing the accuracy of the inventories. Many other studies have used realistic traffic data, namely road-side or on-board observational data obtained at certain road segments, to establish inventories and improve their accuracy (Huo et al., 2009; Wang et al., 2008, 2010; Wang and Xie, 2009; Yao et al., 2013). Although the observed traffic data are helpful for inventory establishment, their limitation is obvious in that large-scale observation for a whole city requires extensive human labor, financial and material resources, which are expensive and time consuming. Moreover, such observation may not provide real-time traffic data, thereby reducing the temporal resolution of the inventories.

Recent developments in image identification technology and other observation detectors, are facilitating easy collection of real-time traffic data from road networks. The extensive implementation of closed circuit televisions and other detection subsystems in the cities helps in the implementation of intelligent transport systems (ITSs) in China (Wu et al., 2009), making it possible to attain real-time traffic data at city scale. Using the traffic data provided by ITSs, many previous studies have successfully developed inventories for different areas in China (Jing et al., 2016; Liu et al., 2018; Zhang et al., 2018). Such studies provided us a new direction for the establishment of bottom-up inventories. The real-time traffic data from the road network could be the most precise input data for on-road emission inventories and could significantly improve the spatial and temporal resolution of the inventories. However, there are still some difficulties in using the ITS data. In some cities, construction of the ITS is not complete yet or has not even been carried out. Moreover, the inconsistency of the data standards leads to an inefficient way of data utilization (Zhang, 2010). Furthermore, the low degree of the data sharing may be the biggest barrier to using traffic data obtained from the ITSs (Huang et al., 2017).

With the help of a high-resolution emission inventory, numerical models can assess the impact of on-road vehicle emissions on the air quality (Huo et al., 2009). The flow and air quality modeling in cities are commonly categorized into four

groups by the length scales, i.e., street scale (~100 m), neighborhood scale (~1 km), city scale (~10 km), and regional scale (~100 km) (Britter and Hanna, 2003). A previous comprehensive literature review on this topic (Zhang et al., 2012) reports that regional-scale chemical transport models (CTMs) have been widely applied to investigate the chemistry and transport of air pollutants from their emission sources. Many studies have successfully applied regional-scale CTMs to investigate the impact of on-road vehicles on the air quality in urban areas in the regional scale (~100 km) (Che et al., 2011; Saikawa et al., 2011; He et al., 2016; Ke et al., 2017). In addition, some researchers have studied street-scale and neighborhood-scale pollutant dispersion and urban air quality by adopting computational fluid dynamics (CFD) models (Fernando et al., 2010; Kim et al., 2012; Kwak et al., 2013; Kwak and Baik, 2014; Park et al., 2015; Zhong et al., 2016; Hang et al., 2017). City-scale (~10 km) CFD modeling, however, usually requires consideration of billions of grids, because a city may include tens of thousands of buildings with high-resolution and complex street networks (Di Sabatino et al., 2008; Ashie and Kono, 2011). Thus, as city-scale CFD simulations are very expensive and time consuming, they are currently rare. Recently, some models have been developed and applied to investigate street-level air quality at the city scale (Davies et al., 2007; Righi et al., 2009; Zhang et al., 2016; Kim et al., 2018) by balancing the requirements of high resolution and low computational cost.

In this direction, the first purpose of this study was to find a new, open-access source of real-time and high-quality traffic data that could serve as the input for developing an on-road emission inventory with high spatial and temporal resolution for cities or urban districts. Guangzhou was selected as the target city for the initial application of this method not only because of the large number of vehicles in use there, but also because of its well-developed ITS which could obtain the traffic information from street networks (Xiong et al., 2010). A Python-based on-road emission model called the Real-time On-road Emission (ROE v1.0) model was developed in this study to utilize these traffic data and establish a bottom-up on-road emission inventory. A street-level chemistry transport model was then used to apply the emission results and study the impact of traffic volume variations on the air quality in the urban districts of Guangzhou.

## 2 Description of the ROE model

### 2.1 Model overview

The ROE model is intended to establish the street-level emission inventories using the emissions of on-road vehicle in the street segment of interest using a bottom-up approach. First, the ROE model collects the real-time traffic information to obtain the traffic volume for each street segment from the ITS. Then, according to the vehicle fleet information, the ROE model calculates the number of vehicles for each vehicle category on each street segment (if available, these data could be obtained from the ITS and need not be calculated by model). Thereafter, the ROE model calculates the emissions for street segments based on the vehicle fleet information, traffic conditions, and environmental conditions. Lastly, the ROE model outputs the results, that is, street-level air quality inventories.

### 2.2 Model structure

The ROE model was developed to calculate on-road vehicle emissions from real-time traffic data. The structure of the model is shown in Figure 1. The model, which has been implemented in Python 3, can be divided into four modules: crawler, preprocessing, emission calculation, and output modules.

(1) The most crucial part of the emission inventory involves obtaining the real-world traffic data. The crawler module is designed for "crawling" the real-time traffic data from the ITS, Internet, or any other data source if the code is updated to match the format of the data source. Moreover, the study area should be set in the module, and if needed (in case the coordinates differ), the coordinate transformation script should be activated. The current version of the ROE model includes the crawler module for the *amap.com* (also called the Gaode map) application (Figure 2), a widely adopted map application in China (additional details are provided in section 2.4). (2) The preprocessing module is used for fitting the time frequency between the data source and the air quality modeling system. Subsequently, the traffic volume data are also calculated from the traffic

speed data in this module if the traffic volume or vehicle fleet information is not available from the data source. Otherwise, the number of vehicles in each category can be used directly for the emission calculation. (3) The emission calculation module uses traffic information from the preprocessing module and information about vehicle fleets to calculate emissions for each street segment using the following equation:

$$E_{s,t} = \sum EF_{s,v} \times V_{v,t} \times L, \tag{1}$$

where $E_{s,t}$ is the emission of pollutant $s$ at time $t$ (g/h), $EF_{s,v}$ is the emission factor of pollutant $s$ for vehicle category $v$ (g/km), $V_{v,t}$ is the traffic volume of the vehicle (i.e., the number of vehicles, veh) category $v$ at time $t$ (veh/h), and $L$ is the length of the street segment (km). The total emission in one specific area is given by the sum of emissions in every street segment within the area. (4) The output module sums up all the information given by the emission calculation module and can be modified to provide all the results produced during the calculation of the emissions. In addition, the model includes a tool that can modify the formats of the emissions, making it possible to provide the on-road emissions to other air quality models.

### 2.3 Emission factors

In this study, nation-wide vehicle emission factors mandated by the Ministry of Ecology and Environment (MEP) of the People's Republic of China were adopted to calculate the on-road    vehicle emissions (MEP, 2014). They are listed in Tables S1 and S2 in the supplementary materials. The emission factors of liquefied petroleum gas (LPG) vehicles were sourced from a previous study conducted in Guangzhou (Zhang et al., 2013). According to the MEP guide, vehicles are classified as one of the following: a light-duty vehicle (LDV), a middle-duty vehicle (MDV), a heavy-duty vehicle (HDV), a light-duty truck (LDT), a middle-duty truck (MDT), a heavy-duty truck (HDT), a motorcycle (MC), a taxi, or a bus. The fuel type is classified as petrol, diesel, or other (such as LPG or natural gas). The emission standard is classified as Pre-China I, China I, China II, China III, China IV, or China V. In addition, the evaporation of petrol was considered during the calculation of the emissions. HC evaporation was also considered as per the details provided in the MEP guidebook (Table S3).

The correction factors involving environmental conditions (e.g., temperature, relative humidity, and altitude) and traffic conditions obtained from the technical guide were considered in the study. They are listed in Tables S4–S10 in the supplementary materials. These correction factors were applied to reduce the effects of uncertainties associated with the emission factors.

To estimate the uncertainties of the emissions factors, the results of previous studies (Zheng et al., 2009a; Zhang et al., 2013, 2016; Tang et al., 2016; Wang et al., 2017) were summarized and compared with the emission factors obtained in this study. These results appear in Figure S1 of the supplementary materials.

In addition, the emission factors can be easily updated once the local emission factors data are available.

### 2.4 Traffic data of floating vehicle

In this study, the traffic speed data of each street segment were obtained from Gaode map. The Gaode map traffic data are quite extensive as it covers over 40 cities in China so far (with most of them being major cities). Based on the GPS and mobile network information, details on vehicle speed and location are collected from the map users' devices while using the map navigation on the road. This aspect saves a considerable amount of human labor and material resources with regard to traffic condition observations. These data are updated in real time and can be used through an open-access application programming interface (API), thus overcoming the barrier of obtaining data. As the data can be updated in real time, the emission data can also be refreshed in real time.

However, the map application cannot provide the traffic volume data directly. Many studies have shown that the traffic volume can be estimated using the average traffic speed based on the relationship between the traffic speed and the volume (Wang, 2003; Kuo and Tang, 2011; Xu et al., 2013; Yao et al., 2013; Hooper et al., 2014; Jing et al., 2016). Many speed–flow models exist for this purpose, and each of them has certain advantages and disadvantages. In this study, the Underwood volume calculation model (Underwood, 1961) was used to retrieve the information on traffic volume because of its history of successful application in China (Jing et al., 2016). The model is described by Eq. (2):

$$V = k_m u \ln \frac{u_f}{u}, \tag{2}$$

where $V$ is the traffic volume at speed $u$ (veh/h), $k_m$ is the traffic density (veh/km), $u$ is the traffic speed (km/h), and $u_f$ is the free speed (km/h). In this study, $k_m$ and $u_f$ are given by fitting the model based on observation data obtained at the roadside and video identification data gained from different road types (Zheng et al., 2009a; Jing et al., 2016; Liu et al., 2018).

5   To calculate the traffic volume on national highways, another speed–flow model, which was previously applied in an observation-based study undertaken in China (Wang, 2003), was used. This model is described as follows:

When the speed limit is 120 km/h,

$$V = -0.611u^2 + 73.320u; \tag{3}$$

when the speed limit is 100 km/h,

$$V = -0.880u^2 + 88.000u; \tag{4}$$

when the speed limit is 80 km/h,

$$V = -1.250u^2 + 100.000u; \tag{5}$$

when the speed limit is 60 km/h,

$$V = -2.000u^2 + 120.000u; \tag{6}$$

15 where $V$ is the traffic volume at speed $u$ (veh/h), and $u$ is the traffic speed (km/h).

   Given Guangzhou's traffic control policies, the whole city is divided into two areas: urban area and suburban (Figure 3). Therefore, the traffic volume is also calculated accordingly (Figure 4). The main traffic control policies in urban areas are as follows: (1) No truck is allowed to enter the urban area during 7:00–9:00 (morning rush hours) and 18:00–20:00 (evening rush hours), (2) no middle- and heavy-duty truck is permitted to enter the urban area during 7:00–22:00, (3) no non-local truck can

20 enter the urban area during 7:00–22:00, and (4) no motorcycle can enter the urban area.

## 2.5 Vehicle fleet information

In this study, the fleet information on each vehicle classification was sourced from the Statistical Yearbook of Guangzhou (Guangzhou Bureau of Statistics, 2017) (Figure 5(a)). The emission standards (Figure 5(b)) and fuel type data (Figure 6) for the vehicles were source from previous studies undertaken in Guangzhou (Zhang et al., 2013, 2015). Due to the lack of the

25 street-level vehicle fleet information, this study used a uniform percentage of emission standard, fuel type and number of vehicles in each category for each segment. The number of each vehicle type was calculated based on the total traffic volume of each street segment and the vehicle fleet percentage. It should be noted that this information could be updated if the street-level fleet information becomes available in the future.

## 3 Description of the street-level air quality model

30 To evaluate the impact of on-road emissions on air quality at the street level in Guangzhou, an air quality model called the Model of Urban Network of Intersecting Canyons and Highways (MUNICH) was employed in this study with the on-road emission results from the ROE model. MUNICH is a street-network CTM that includes street-canyon and street-intersection components in the model (Kim et al., 2018).

   In this study, the Weather Research and Forecasting (WRF) model (version 3.7.1) (Skamarock et al., 2008) was used to

35 provide the meteorological data (wind profile, boundary layer height, and friction velocity) for the modeling. The WRF simulation was conducted with four nested domains at resolutions of 27 km, 9 km, 3 km, and 1 km (Figure 7a). The physical scheme is listed in Table 1.

   In MUNICH, the CB05 chemical kinetic mechanism (Yarwood et al., 2005) was used to simulate the photochemical reactions at the street level in an urban street network. For the MUNICH run, the model was applied to simulate pollutant

40 dispersion in Tianhe District, which serves as the Central Business District (CBD) of Guangzhou. The district is characterized by significant diurnal traffic variation compared with other districts in urban areas. The simulation area comprised 31 main

street segments selected to simulate the variation in pollutant concentrations, because continuous traffic data existed for these street segments during the simulation period, which were representative of the street network.

The urban morphology data for the building height were obtained from the World Urban Database and Access Portal Tools (WUDAPT) dataset (Ching et al., 2018). The street data were sourced from the OpenStreetMap dataset (https://www.openstreetmap.org/). The street length data were calculated directly from the locations of the start and end intersections of each street segment. Data on the street width were retrieved from the feature class of the road, and the width of each lane was assumed to be 3.5 m.

The simulation period of the study spanned from April 28[th], 2018 to May 2[nd], 2018, which included a Chinese national holiday from April 29[th], 2018 to May 1[st], 2018. Significant traffic volume change exists between the holidays and non-holidays. This simulation period covered holidays and non-holidays, which was helpful to investigate the impact of traffic volume variations on air quality. Another 3-day simulation period was conducted before this period to spin up the model.

For modeling evaluation and background concentrations, the observational concentration data for $NO_2$ and $O_3$ were obtained from the Guangzhou environmental monitoring sites network. $NO_2$ concentrations were measured with a chemiluminescence instrument (Model 42i, Thermo Scientific) and $O_3$ was measured by a UV photometric analyzer (Model 49i, Thermo Scientific). The minimum detection limit (3S/N) of the analyzer was 0.4 ppbV (approximately 0.8 $\mu g/m^3$) for $NO_2$ and 1.0 ppbV (approximately 2.0 $\mu g/m^3$) for $O_3$. The total measurement uncertainty of these two instruments was estimated to be approximately 5% (Zhang et al., 2014).

Two monitoring sites, Tiyuxi (TYX) site and YangJi (YJ) site, were selected for this study (Figure 7c). The observational data from TYX were used for modeling evaluation because TYX locates inside the simulation area, and thus these data which could be used for comparison with the model results. In addition, YJ is located near but not within the simulation street network. The observational data from YJ could be used as the background concentration data for the modeling. Due to the lack of NO observational data, the concentration ratio of $NO_2$ to NO was assumed as 4:1 in this study.

## 4 Application of the ROE model to Guangzhou

### 4.1 On-road emission inventory from the ROE model

#### 4.1.1 Overview of the emission inventory

Using the high-resolution spatial and temporal traffic data from the map application, the emission inventory of on-road vehicles from the ROE model was established for this study. Table 2 shows the annual emissions from vehicles in Guangzhou city compared with two other gridded emission inventories in China: the MEIC model (http://meicmodel.org/) and a PRD region local emission inventory (Zheng et al., 2009b). These two emission inventories used the top-down method to establish on-road emission inventories. Unlike the bottom-up method used in this study, these two inventories first calculated the total emissions based on the VKT data of vehicle categories. In the MEIC inventory, the total number of vehicles was obtained from the relationship between total vehicle ownership and economic development (Zheng et al., 2014), while the PRD inventory acquired information on the number of vehicles from the city-level statistics Yearbook. Then, the spatial distribution of these two inventories was established based on the road network density.

Given the shorter total road length and traffic control policies in urban areas (Figure 3), the urban on-road emissions of CO, $NO_x$, HC, $PM_{2.5}$, and $PM_{10}$ comprised only 13.1%, 8.8%, 12.7%, 8.2%, and 9.1% of the total on-road emissions, respectively, suggesting that the suburban areas are the dominant contributor of on-road emissions in Guangzhou.

In general, the difference between the amounts of $PM_{2.5}$ and $PM_{10}$ was smaller than that for other gaseous emissions among different inventories. This was because the uncertainty of particulate matter emission factors was lower than the corresponding values of the other gaseous emissions, which led to the large difference for the gaseous emissions and the smaller differences for $PM_{2.5}$ and $PM_{10}$. For $NO_x$ emissions, however, this study showed a higher $NO_x$ estimate than that in the other two inventories. One of the reasons for the higher $NO_x$ estimate may be the application of the updated LPG bus emission

factors in this study. Based on a previous local emission factor study, the $NO_x$ emission factor of an LPG-fueled bus is 1.7 times that of a diesel-fueled bus in Guangzhou (Zhang et al., 2013). The results in Figure 8 show that the $NO_x$ emissions attributable to buses in urban and suburban areas were 20.5% and 10.8% of the total $NO_x$ emissions, respectively, showing that the LPG-fueled buses may be responsible for higher $NO_x$ estimates in this study compared to those in the other two inventories.

As shown in Table 3, the emission contribution of local roads in urban areas is the highest component because of the total length of the local roads, which is 5.4 times and 4.8 times that of highways and arterial roads in urban areas, respectively. Although the total length of the highways is shorter, the traffic volume on the highway is much higher than that on the local roads (Figure 4), thus causing the highest contribution of emissions from the suburban areas. Moreover, the emission contributions from urban and suburban areas differ on weekdays and weekends. In urban areas, the daily total weekday and

weekend emissions are 129.94 Mg/d and 118.29 Mg/d of CO, 30.15 Mg/d and 27.71 Mg/d of $NO_x$, 14.74 Mg/d and 13.40 Mg/d of HC, 1.27 Mg/d and 1.16 Mg/d of $PM_{2.5}$, and 1.41 Mg/d and 1.29 Mg/d of $PM_{10}$, respectively. In suburban areas, the total weekday and weekend emissions are 873.97 Mg/d and 758.41 Mg/d of CO, 315.10 Mg/d and 267.91 Mg/d of $NO_x$, 102.46 Mg/d and 88.22 Mg/d of HC, 13.01 Mg/d and 10.98 Mg/d of $PM_{2.5}$, and 14.45 Mg/d and 12.19 Mg/d of $PM_{10}$, respectively. The total respective emissions of CO, $NO_x$, HC, $PM_{2.5}$, and $PM_{10}$ on a weekday are 114.5%, 116.8%, 115.3%, 117.6%, and

117.7% of the values on a weekend, respectively.

### 4.1.2 Spatial distribution of emissions

Due to the vehicular activities, the spatial distribution of on-road emissions was consistent with the structure of the street network. For a better description of this spatial distribution, the emissions were mapped onto a 1-km-resolution fishnet and the total emissions of one grid cell were the sum of all on-road emissions from within the same grid cell. The spatial distribution

of each pollutant is shown in Figure 9. Overall, the high-value grid cells were generally located along the highways. In suburban areas, high-value areas located away from the highways and arterial roads normally denoted suburban town centers that had more local roads and higher traffic volume density. In urban areas, the high-value areas were more closely related to the densities of the urban local roads. The emission hotspots were less prominent in urban areas than in suburban areas due to the strict traffic control policies in urban area. The spatial distribution indicated that the next effective control on-road emissions

policy should pay more attention to the control of vehicles in suburban areas.

    Moreover, the spatial distributions of these three emission inventories were compared in this study. Figure 10 shows the distributions of CO from the three different inventories. The results of both MEIC-2016 and PRD-2015 showed the urban areas as emission hotspots. However, the results from the ROE model were much lower for such areas. This may be due to the fact that the ROE model considers the traffic control policies, while the other two inventories do not. In suburban town centers,

especially in the eastern and southern parts of Guangzhou, all three inventories showed the same results, namely that these areas were large contributors of on-road emissions. Notably, highways and arterial roads also contributed high emissions in all three inventories.

### 4.1.3 Emission contributions of vehicles by their classification

The emission contributions of different vehicle classifications in the urban and suburban areas are shown in Figure 8. As LDVs

accounted for the largest number, their emission contribution comprised the dominant proportion of total emissions in urban areas for each pollutant. The contribution percentages of CO, HC, $NO_x$, $PM_{2.5}$, and $PM_{10}$ were 80.9%, 84.1%, 26.4%, 38.3%, and 38.2%, respectively. HDVs were the second largest contributor to on-road emissions, the relevant percentages being 5.8%, 2.9%, 30.3%, 35.2%, and 35.2% for CO, HC, $NO_x$, $PM_{2.5}$, and $PM_{10}$, respectively. As for the buses, except for the contribution of $NO_x$, which accounted for 20.5% of the total emissions mentioned above, the proportions of the other pollutants were less

than 2% because of the use of LPG as the fuel. In the case of trucks, the total contribution of LDTs, MDTs, and HDTs were 10.3%, 9.3%, 21.2%, 23.3%, and 23.3% for CO, HC, $NO_x$, $PM_{2.5}$, and $PM_{10}$, respectively, considering the traffic control policies in the urban areas. The contribution of taxi was less than 1% because of the small number of taxis and their use of LPG. In suburban areas, the LDVs were the dominant contributor of CO and HC emissions because of their high numbers. For

$NO_x$, $PM_{2.5}$, and $PM_{10}$, however, the HDT provided the largest contribution, at 36.5%, 43.2%, and 43.3%, respectively. Moreover, LDVs, HDVs, and buses were important contributors of $NO_x$, at 19.4%, 17.4%, and 10.8%, respectively. Regarding particulate matter, the respective percentages of emissions (for both $PM_{2.5}$ and $PM_{10}$) owing to LDVs, HDVs, and LDTs were 19.7%, 20.5%, and 9.0%, suggesting that these vehicles were also important sources of both $PM_{2.5}$ and $PM_{10}$.

### 4.2 Application of the ROE model's results to the street-level air quality model

#### 4.2.1 Modeling performance in Guangzhou urban area

During the simulation period, the model results were evaluated for the TYX observation site located within the street network. The on-road emissions were provided by the ROE model, as discussed previously. Street segments to which high $NO_x$ emission values were attributed were also responsible for high HC emissions because of the positive relationship between traffic volume and on-road emissions as shown in Figure 11.

The time series for the simulated $NO_2$ and $O_3$ concentrations within the street network were compared with the observed concentrations (Figure 12). As the results show, daytime $NO_2$ concentrations were overestimated while nighttime concentrations were underestimated during the simulation period. The $O_3$ concentrations, however, were underpredicted during daytime and overpredicted at nighttime. Several modeling sensitivity cases were analyzed to identify what factors may have affected the model simulation. The sensitivity analysis results are provided in the supplementary materials section S3. Typically, the overestimated background concentrations of $NO_2$ and $O_3$ were attributed as the reason for the overprediction of the daytime $NO_2$ and nighttime $O_3$ concentrations, respectively. The underestimated NO titration was the other main reason for the overprediction of $O_3$ and the underprediction of $NO_2$ concentrations at night. Due to the only consideration of on-road emission in the simulation street network, daytime $O_3$ concentrations were underpredicted in the results.

Moreover, the performance statistics for $NO_2$ and $O_3$ are shown in Table 4. Here, the statistical measures of the observation (OBS) mean, simulation (SIM) mean, mean bias (MB), normalized mean bias (NMB), normalized mean error (NME), mean relative bias (MRB), mean relative error (MRE), root mean squared error (RMSE), and the correlation coefficient (CORR) were used to validate the model. The NMB, NME, and CORR values of $NO_2$ and $O_3$ in this study were within the recommended ranges in the MEP Technical Guide for Air Quality Model Selection (MEP, 2012). These recommended values were -40% < NMB < 50%, NME < 80%, and $R^2$ > 0.3 for $NO_2$, -15% < NMB < 15%, NME < 35%, and $R^2$ > 0.4 for $O_3$. Additionally, the values obtained in this study fell within the range of those reported by other modeling studies in Guangzhou; the NMB, NME, and RMSE values for simulated urban $NO_2$ in Guangzhou ranged from -27.5% to -6%, 29.2% to 53.0%, and 16 to 37.3, respectively, and the corresponding ranges for $O_3$ were -21.2% to 20.0%, 38.2% to 98%, and 9.4 to 40.1 (Che et al., 2011; Fan et al., 2015; Wang et al., 2016). Overall, the model showed good simulation performance and can be applied to future studies investigating the impact of on-road vehicles on air quality.

#### 4.2.2 Impact of traffic volume variations on air quality

To investigate how traffic volume change affects air quality at the street level, a Chinese national holiday was chosen as the target simulation period for the modeling. Figure 13 shows the diurnal variation in the traffic volume during the national holiday, normal weekday, and normal weekend before and after the holiday in the simulation street network. On the normal weekday, two typical rush hour trends appeared during the 8:00–10:00 and 17:00–19:00 (although April 28[th] was a Saturday, it was a normal workday to compensate for the holiday). For the normal weekend and the national holiday, the peak in traffic volume was noted between 14:00 and 16:00 and no rush hour peak occurred on these days. At nighttime, not much difference was noted for the traffic volumes on the normal weekday, normal weekend, and national holiday, especially after midnight. However, the higher traffic volume between 21:00 and 23:00 on April 28[th] at night may have been caused by people traveling out of the city before the national holiday (e.g., returning home across the city or traveling to other places).

Three sensitivity cases were carried out to study the impact of traffic volume change on the air quality in urban areas: (1) in the national holiday case, wherein the on-road emissions between April 29$^{th}$ and May 1$^{st}$ were regarded as the original emissions during the simulation period(this represents the base case), (2) in the normal weekday case, diurnal on-road emissions for three national holidays were replaced by the emissions of April 28$^{th}$, and (3) in the normal weekend case, the national holiday period emissions were replaced by the diurnal on-road emissions of May 5$^{th}$. The diurnal variations in NO$_x$ and O$_3$ in the three cases are shown in Figure 14. During 0:00–5:00, because of similar traffic volume, there were no large differences in the NO$_x$ and O$_3$ concentrations during this time. Due to the morning rush hour, the NO$_x$ concentrations for the normal weekday case were much higher than those for the national holiday case in the morning. As shown in Table 5, the NO$_x$ concentrations were 12.0–26.5% higher for the normal weekday case during this time. In the normal weekend case too, the NO$_x$ concentrations simultaneously increased by 9.1% compared to those on the national holiday in the morning. This increase was caused by people traveling for normal weekend engagements. In the afternoon, however, the difference between the NO$_x$ concentrations was less than 10% due to the rising traffic volume on the national holiday. During the evening rush hour, although the traffic volume on the normal weekday was 1.3 times that on the national holiday, the maximum difference between the NO$_x$ concentrations was only 7.3%. This shows that the variations in NO$_x$ concentrations were affected to a greater extent by the background concentrations (i.e., boundary conditions) in the evening.

Compared with the national holiday case, the O$_3$ concentrations were much lower in the normal weekday case. In the afternoon, as shown in Table 6, when photochemical reactions are more prevalent, the national holiday O$_3$ concentrations exceeded those on normal weekdays and weekends by 13.9% and 10.6%, respectively. This is because the simulation street network in the urban areas is in the VOC-sensitive regime (Ye et al., 2016). The O$_3$ concentrations were positively correlated with the VOC emissions. As the NO$_x$ emissions were higher than the VOC emissions, the reduction in the NO$_x$ emissions was also much higher than in the VOC emissions when the number of vehicles decreased on the national holiday. The larger NO$_x$ emission reduction led to a higher VOCs-to-NO$_x$ emission ratio, which resulted in a higher O$_3$ concentration during the national holiday (Sanford and He, 2002).

## 5 Discussion and conclusions

Using real-world traffic information, the Real-time On-road Emission (ROE v1.0) model can provide real-time and high-resolution emission inventories for regional or street-level air quality models in China. The results show that the ROE model can simulate the emissions of CO, NO$_x$, HC, PM$_{2.5}$, PM$_{10}$ and any other pollutant provided the relevant emission factors are included in the model. (This aspect will be updated in subsequent releases.) As it uses the bottom-up method, the ROE model facilitates the calculation of the emissions in each street segment.

In this study, the traffic information of Guangzhou was obtained from the Gaode map, the data for which are collected from map users while they are driving. The geographic and speed information were sourced from the map users' GPS devices and can be used through the map API. Using the ROE model and fully considering the traffic control policies of Guangzhou city, the annual total on-road emissions of CO, NO$_x$, HC, PM$_{2.5}$, and PM$_{10}$ were modeled to be $35.22 \times 10^4$ Mg/yr, $12.05 \times 10^4$ Mg/yr, $4.10 \times 10^4$ Mg/yr, $0.49 \times 10^4$ Mg/yr, and $0.55 \times 10^4$ Mg/yr, respectively. Spatial distribution analysis showed that hotspots of on-road emissions were situated along the highways and suburban town centers. The comparison of spatial distribution between the ROE model's results and those of two other inventories showed that the ROE model provided had lower urban emissions as it considered the traffic control polices. However, it should be noted that this comparison was only preliminary. The spatial resolutions of the three inventories are inconsistent in this study. Moreover, due to the lack of temporal information about the other two emission inventories, a comparison of the temporal difference could not be conducted. Future studies should focus on improving the accuracy of such comparisons.

Owing to the number of vehicles and their respective distributions, LDVs constituted the dominant source of on-road emissions in Guangzhou. In suburban areas, however, HDTs were the highest contributors of NO$_x$, PM$_{2.5}$, and PM$_{10}$. Daily emissions of CO, NO$_x$, HC, PM$_{2.5}$, and PM$_{10}$ on a weekday were found to be 14.5%, 16.8%, 15.3%, 17.6, and 17.7% higher

than the daily emissions on a weekend, respectively. However, due to the lack of street-level vehicle fleet information, this study applied a city-level average uniform percentage for every street segment. This may increase the uncertainty of the inventory, but this aspect could be improved upon provided additional data become available in the future. Given the high spatial and temporal resolutions of the emission inventory of the ROE model, three sensitivity cases were analyzed to study the effect of vehicular on-road emissions on urban street-level air quality. On a national holiday, $NO_x$ concentrations were 12.0–26.5% less than those on a normal weekday as no morning rush hours occurred on holidays. Moreover, compared with the normal weekend, the $NO_x$ concentrations on a national holiday also show a decrease of 9.1% in the peak value in the morning. However, the reduction in the $NO_x$ concentrations in the afternoon was smaller than that in the morning, suggesting that the transportation of $NO_x$ from the surrounding areas was the main reason for the variation in the afternoon $NO_x$ concentrations. In addition, as the simulation street network lies in the VOC-sensitive regime, the lower traffic on a national holiday and a normal weekend caused the $NO_x$ and VOC emissions to be lower than those on a normal weekday. However, the reductions in $NO_x$ were higher than the decrease in VOC emissions, which led to a higher VOCs-to-$NO_x$ emission ratio and $O_3$ concentrations on holidays and normal weekends. In this study, only 31 main street segments were selected to study the impact of a holiday on air quality in a certain urban area of Guangzhou. Additional investigations are required to understand the variations in street-level air quality in urban or suburban area of a megacity. The results of the ROE model showed that the suburban town centers of Guangzhou served as emission hotspots. These areas had relatively higher emissions than the other suburban areas and less stringent control policies than the urban area, which suffers from more serious air quality problems.

In general, the ROE model could provide a high-resolution on-road emission inventory when the real-time traffic information and emission factors were fed into the model. It is worth noting that the ROE model is highly dependent on the ITS traffic data. For economically underdeveloped cities, this aspect may pose a barrier against the use of the ROE model. In addition, China is promoting the CHINA VI emission standards for on-road vehicles. The ROE model only considers Pre-CHINA I to CHINA V currently. Thus, the model will be updated in the near future to include the CHINA VI emission standards.

Recently studies had shown that traffic forecasting models are effective within cities (Min et al., 2009; Cortez et al., 2012; Vlahogianni et al., 2014). These models allow one to obtain predicted traffic-based on-road emissions. Combined with the meteorological forecasting systems and regional air quality forecasting systems, which provide the meteorological and background concentration predictions, respectively, street-level air quality models could be used for street-level air quality forecasting as well.

In summary, the newly developed ROE model was confirmed to be effective for analyzing real-time city-scale traffic emissions and performing high-resolution air quality assessments in the street networks of Guangzhou city. The methodologies presented in this work can be further extended to more typical cities or urban districts in China or other countries.

*Author contribution.* Luolin Wu and Xuemei Wang designed the experiments. Luolin Wu developed the model code and performed the simulations. Ming Chang organized and visualized the data. Jinpu Zhang collected and organized the observational data. Luolin Wu and Jian Hang prepared the manuscript with contributions from all co-authors. Liqing Wu organized the results of model sensitivity cases.

*Code availability.* The python source code of the ROE v1.0 model and examples are available on Github (https://github.com/vnuni23/ROE) and Zenodo (https://doi.org/10.5281/zenodo.3264859). More information and help are also available by contacting the authors.

*Competing interests.* The authors declare that they have no conflict of interest.

*Acknowledgements.* This work was supported by National Nature Science Fund for Distinguished Young Scholars (41425020), the National Key Research and Development Program of China (2016YFC0202206) and the State Key Program of National Natural Science Foundation of China (91644215), as well as National Natural Science Foundation-- Outstanding Youth Foundation (41622502).

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

**Table 1. Physical parameterization configurations for WRF v3.7.1 model**

| Physical parameterizations | |
|---|---|
| Microphysics Scheme | Morrison (2 moments) (Morrison et al., 2009) |
| Land-surface Scheme | Pleim-Xiu (Xiu and Pleim, 2001) |
| Cumulus Scheme | Kain-Fristsch (Kain, 2004) |
| Longwave Radiation Scheme | Rapid Radiative Transfer Model (RRTM) (Mlawer et al., 1997) |
| Shortwave Radiation Scheme | Dudhia (Dudhia, 1989) |
| Boundary-layer Scheme | Asymmetric Convective Model version 2 (ACM2) (Pleim, 2007) |
| Urban Surface Scheme | Urban Canopy Model (UCM) (Chen et al., 2011) |

**Table 2. Annual on-road emissions in Guangzhou (unit: $10^4$ Mg/yr)**

| | | CO | NO$_x$ | HC | PM$_{2.5}$ | PM$_{10}$ |
|---|---|---|---|---|---|---|
| | Urban | 4.61 | 1.07 | 0.52 | 0.04 | 0.05 |
| This study | Suburban | 30.61 | 10.98 | 3.58 | 0.45 | 0.50 |
| | Total | 35.22 | 12.05 | 4.10 | 0.49 | 0.55 |
| MEIC-2016 | (Gridded) | 43.56 | 8.45 | 9.26 | 0.46 | 0.47 |
| PRD-2015 | (Gridded) | 28.89 | 6.99 | 4.65 | 0.52 | 0.52 |

**Table 3. Daily emissions on different road type in urban and suburban area (unit: Mg/day)**

| | | Road type | Length(km) | CO | NO$_x$ | HC | PM2.5 | PM$_{10}$ |
|---|---|---|---|---|---|---|---|---|
| weekday | urban | highway | 301.87 | 9.71 | 3.15 | 1.02 | 0.11 | 0.12 |
| | | artery | 337.19 | 17.24 | 4.95 | 1.88 | 0.19 | 0.21 |
| | | local | 1629.92 | 102.99 | 22.05 | 11.84 | 0.97 | 1.08 |
| | suburban | highway | 2316.73 | 417.49 | 168.29 | 45.51 | 6.50 | 7.22 |
| | | artery | 747.63 | 61.12 | 26.54 | 7.24 | 1.11 | 1.23 |
| | | local | 8867.69 | 395.36 | 120.27 | 49.71 | 5.40 | 6.00 |
| weekend | urban | highway | 301.87 | 7.47 | 2.34 | 0.79 | 0.08 | 0.09 |
| | | artery | 337.19 | 13.20 | 4.23 | 1.40 | 0.15 | 0.17 |
| | | local | 1629.92 | 97.62 | 21.14 | 11.21 | 0.93 | 1.03 |
| | suburban | highway | 2316.73 | 428.30 | 156.78 | 47.14 | 6.07 | 6.74 |

| | artery | 747.63 | 59.20 | 26.56 | 6.99 | 1.10 | 1.22 |
| | local | 8867.69 | 270.91 | 84.57 | 34.09 | 3.81 | 4.23 |

**Table 4 The performance statistics for NO$_2$ and O$_3$ in modeling (unit: μg/m$^3$)**

| | Mean | | MB[c] | NMB[d] | NME[e] | MRB[f] | MRE[g] | RMSE[h] | CORR[i] |
| | OBS[a] | SIM[b] | | | | | | | |
|---|---|---|---|---|---|---|---|---|---|
| NO$_2$ | 30.8 | 35.4 | 4.7 | 15.2% | 68.8% | 3.0% | 3.2% | 25.7 | 0.90 |
| O$_3$ | 60.9 | 59.3 | -1.6 | -2.7% | 24.3% | <0.1% | 0.3% | 18.7 | 0.80 |

[a] OBS (Observation). [b] SIM (Simulation). [c] MB (Mean Bias). [d] NMB (Normalized Mean Bias). [e] NME (Normalized Mean Error). [f] MRB (Mean Relative Bias). [g] MRE (Mean Relative Error). [h] RMSE (Root Mean Squared Error). [i] CORR (correlation coefficient).

**Table 5. Daytime percentage difference of NO$_x$ compared to National holiday case**

| time | 6:00 | 7:00 | 8:00 | 9:00 | 10:00 | 11:00 | 12:00 | 13:00 | 14:00 | 15:00 | 16:00 | 17:00 | 18:00 | 19:00 | 20:00 |
|---|---|---|---|---|---|---|---|---|---|---|---|---|---|---|---|
| normal weekday | 12.7 | 21.7 | 16.8 | 26.5 | 14.7 | 12.0 | 4.9 | 0.6 | 8.6 | 2.2 | 0.7 | 0.2 | 7.3 | 5.9 | 7.1 |
| normal weekend | -4.4 | 0.1 | 9.1 | 6.7 | 0.2 | 7.0 | 1.2 | 2.6 | 6.2 | 0.8 | -0.6 | -0.9 | 2.1 | -5.7 | 4.9 |

**Table 6. Daytime percentage difference of O$_3$ compared to National holiday case**

| time | 6:00 | 7:00 | 8:00 | 9:00 | 10:00 | 11:00 | 12:00 | 13:00 | 14:00 | 15:00 | 16:00 | 17:00 | 18:00 | 19:00 | 20:00 |
|---|---|---|---|---|---|---|---|---|---|---|---|---|---|---|---|
| normal weekday | -4.5 | -15.7 | -52.8 | -48.9 | -37.5 | -25.9 | -15.6 | 0.2 | -7.9 | -13.9 | -7.4 | -11.1 | -46.3 | -38.4 | -32.3 |
| normal weekend | 2.9 | 6.3 | -2.6 | -4.9 | -15.0 | 0.5 | -4.0 | -1.6 | -5.7 | -10.6 | -0.4 | 12.4 | -15.3 | -0.1 | 3.7 |

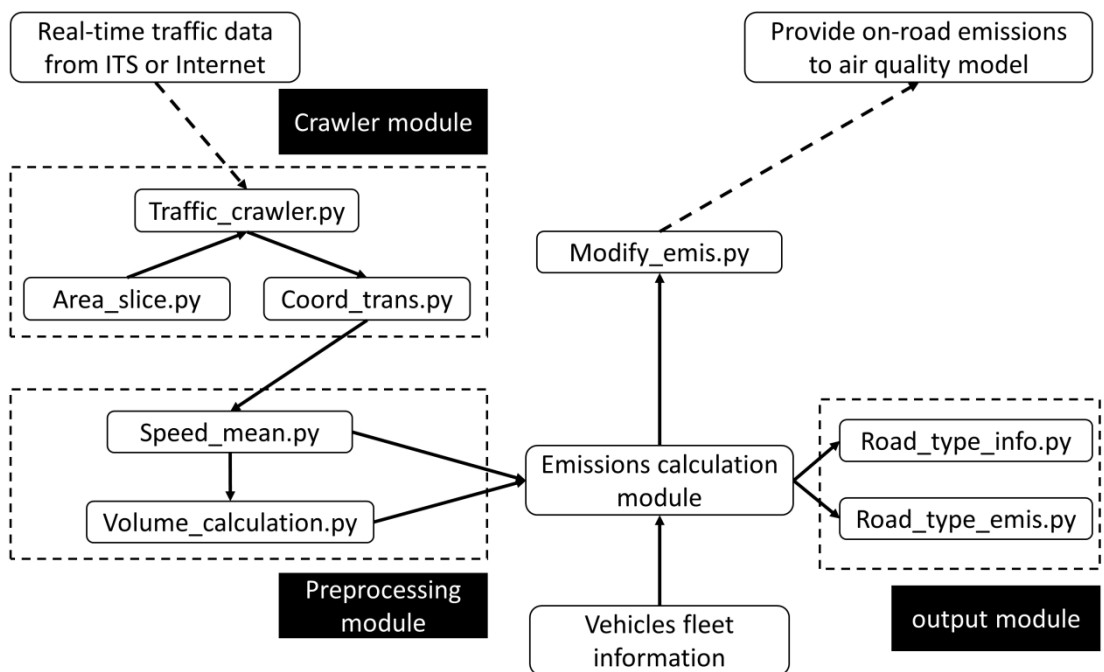

**Figure 1. The structure of ROE model.**

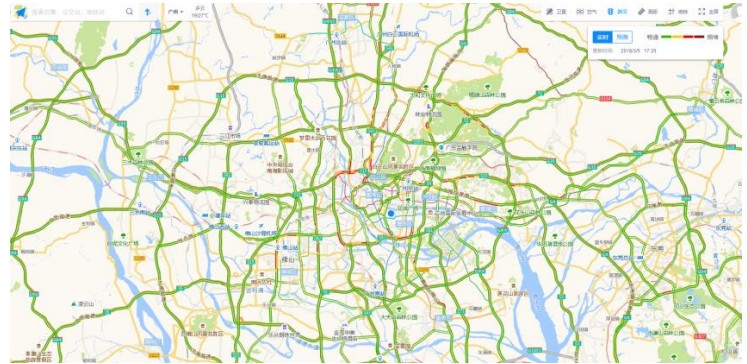

5    **Figure 2. Traffic information from Gaode map.**

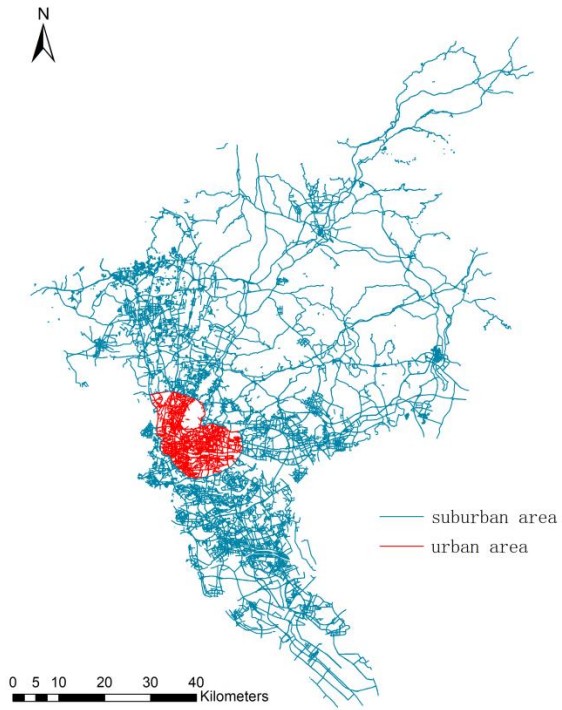

**Figure 3. Traffic control area.**

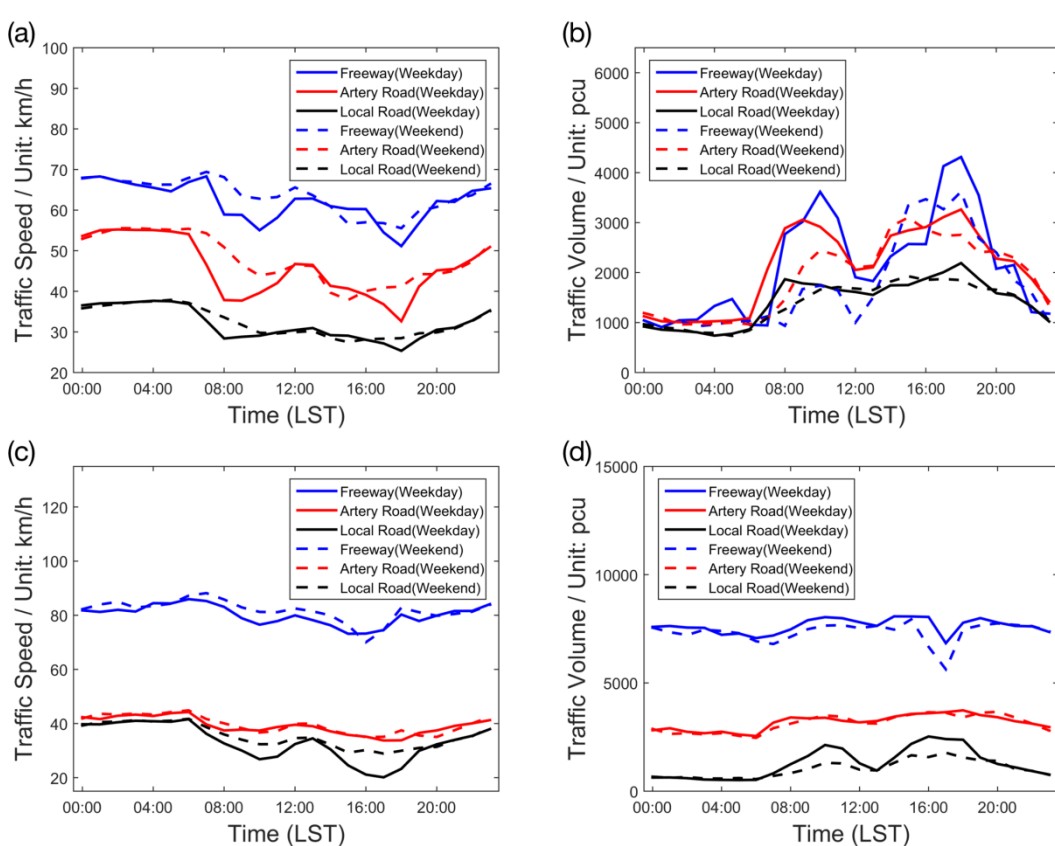

**Figure 4. Diurnal variation of average traffic speed and traffic volume in (a, b) urban area and (c, d) suburban area during weekday and weekend.**

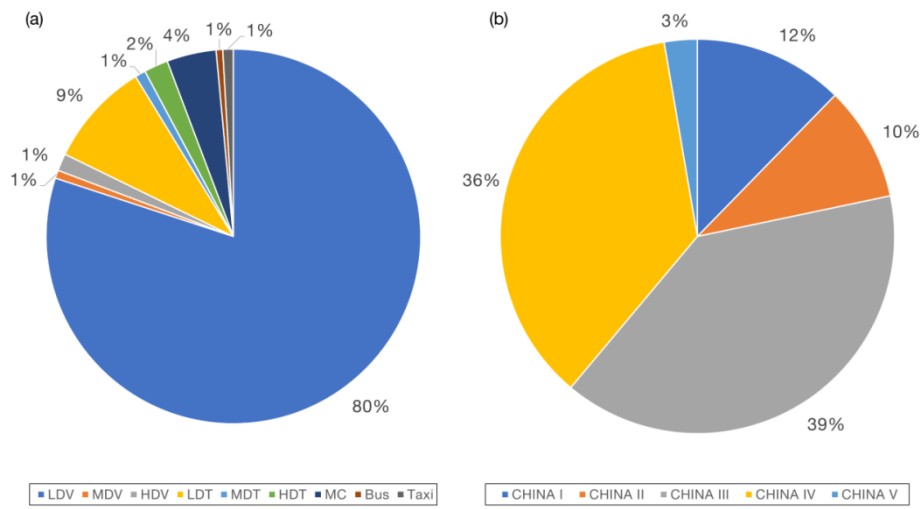

**Figure 5. The percentage of (a) vehicle classification and (b) emission standard.**

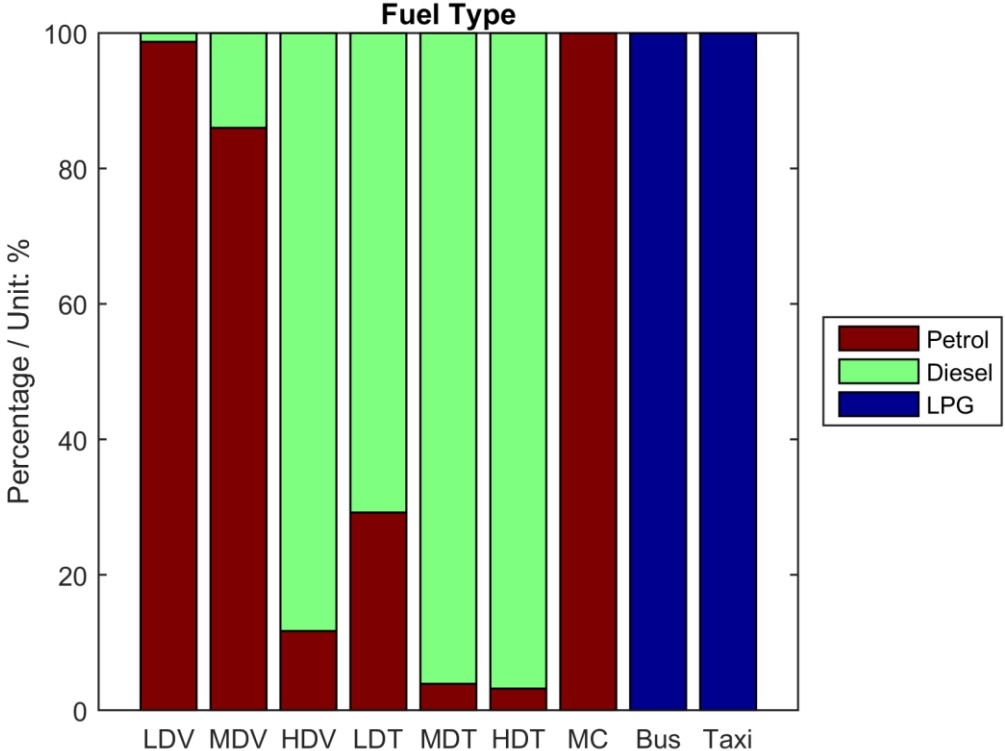

Figure 6. Fuel type percentage of each vehicle classification.

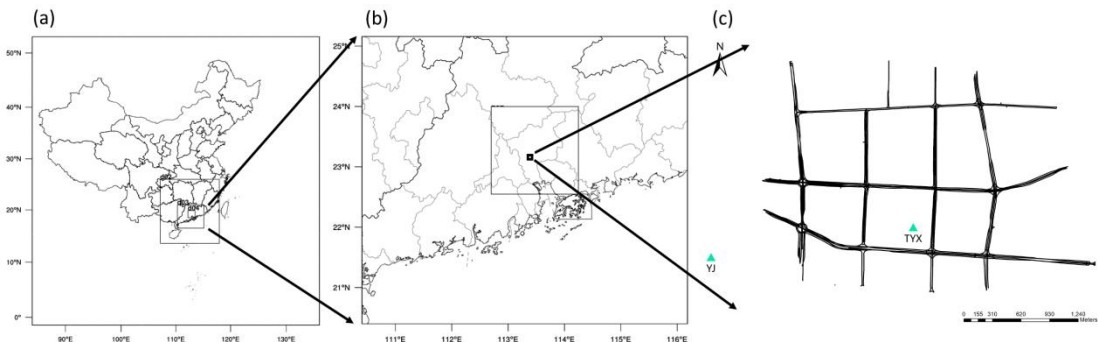

**Figure 7. Simulation domain from regional scale to street-level scale: (a) four-nested simulation for WRF; (b) domains 3 and 4 covering the Pearl River Delta Region and Guangzhou city, the innermost box corresponds to the Tianhe District; (c) 31 street segments and two observation sites (triangle) within the MUNICH study domain.**

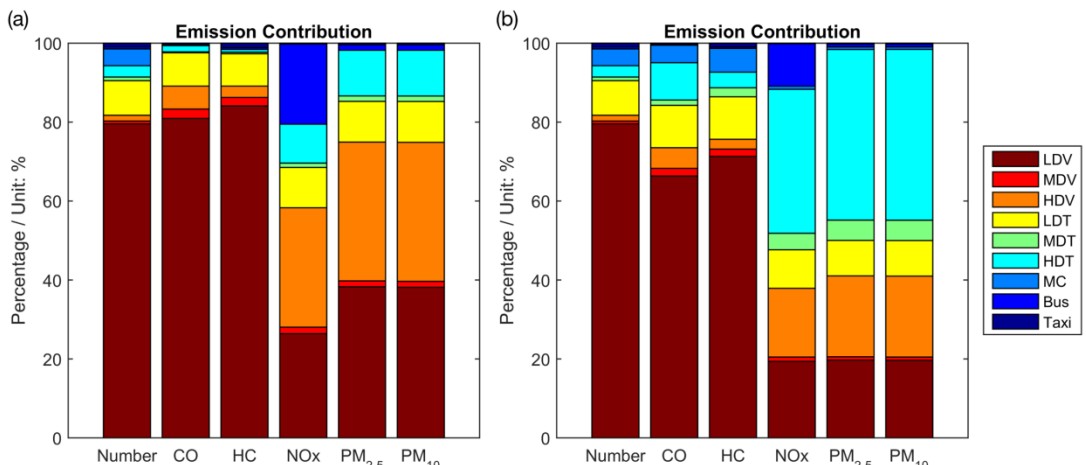

**Figure 8. Emission contribution of each vehicle classification in (a) urban area and (b) suburban area.**

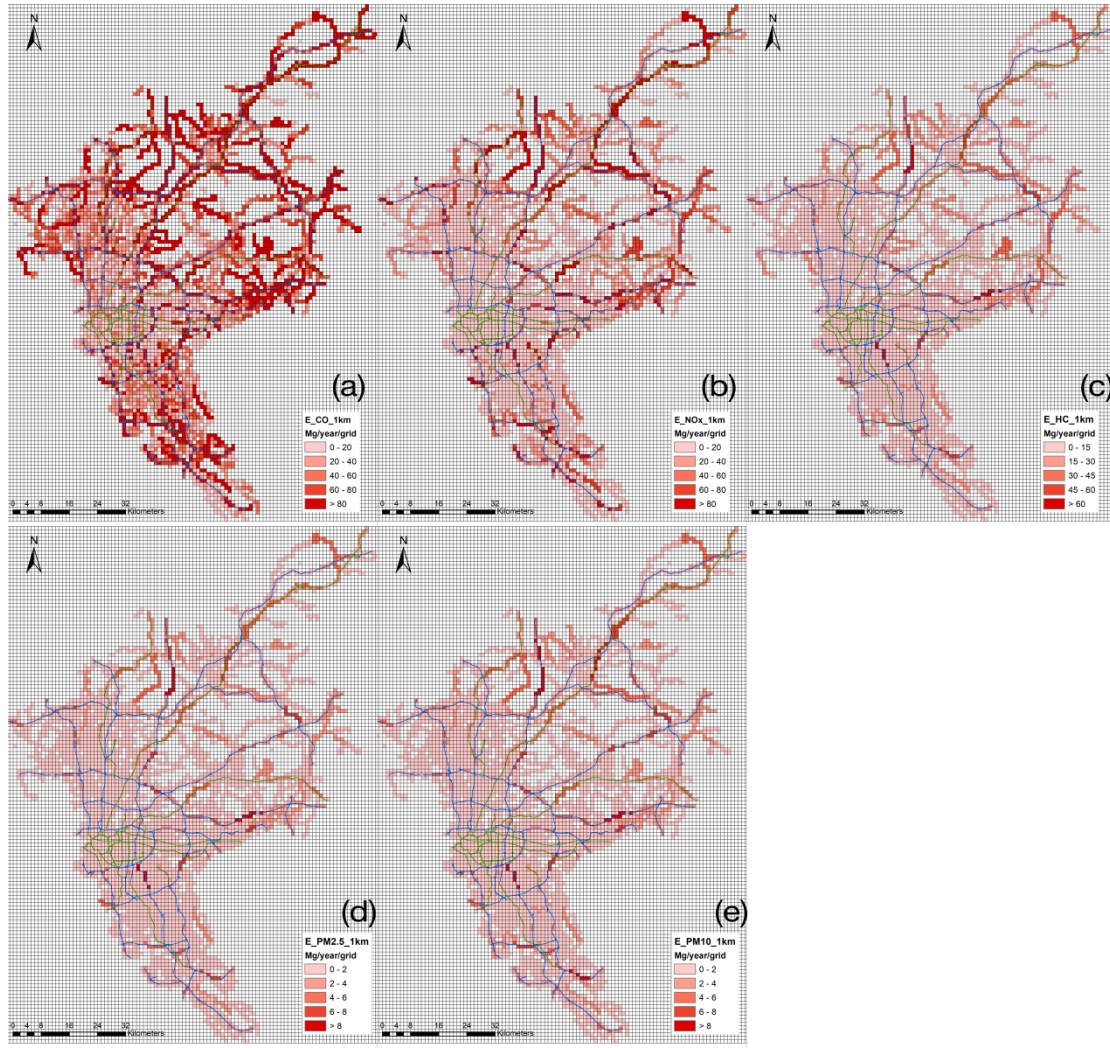

**Figure 9. Spatial distribution of (a) CO, (b) NOₓ, (c) HC, (d) PM₂.₅, and (e) PM₁₀ from the on-road emissions in Guangzhou (blue lines: highways; green lines: arterial roads; local roads are not shown).**

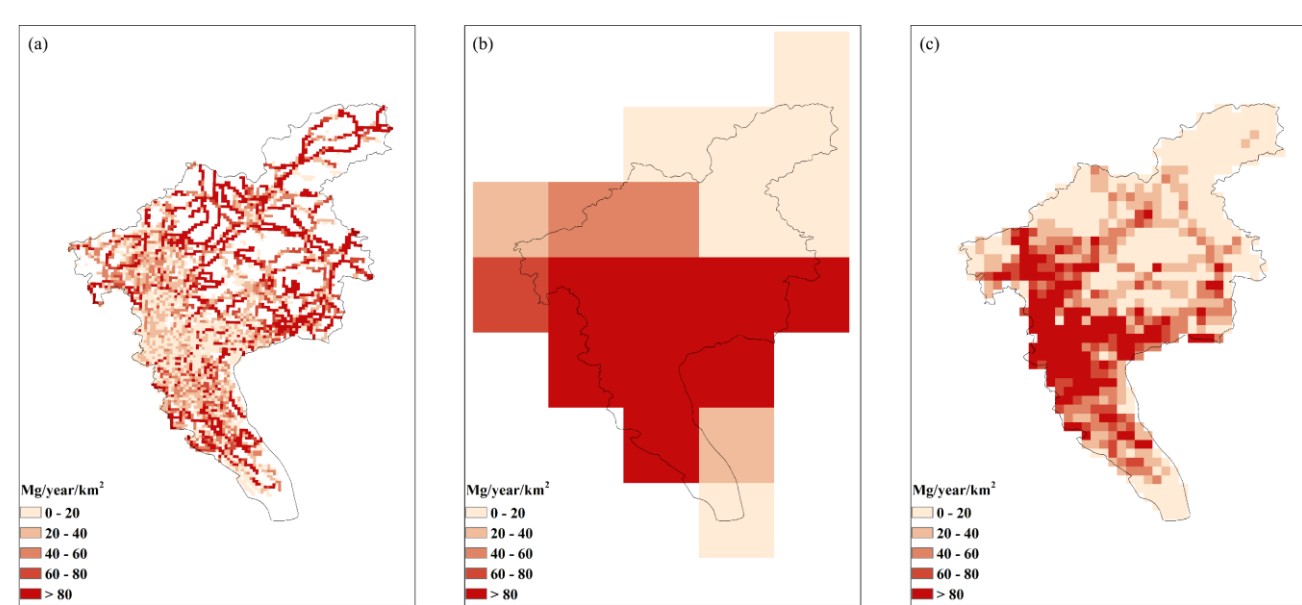

**Figure 10. Spatial distribution of CO from (a) ROE model, (b) MEIC-2016 and (c) PRD-2015 in Guangzhou.**

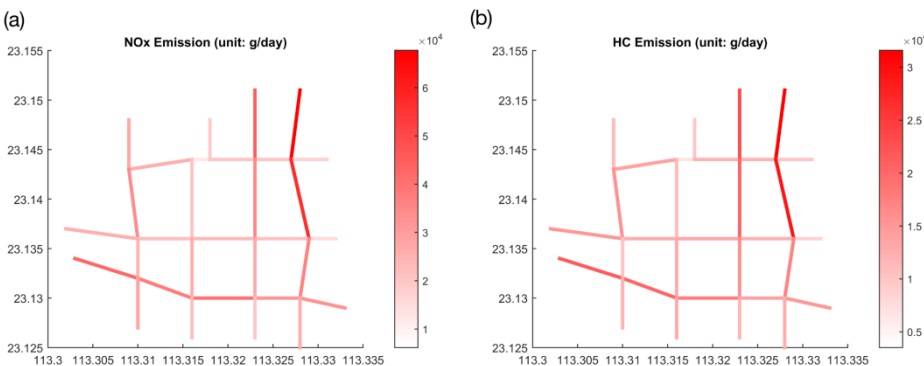

**Figure 11. The spatial distribution of weekday (a) NOₓ and (b) HC emission in the simulated street network.**

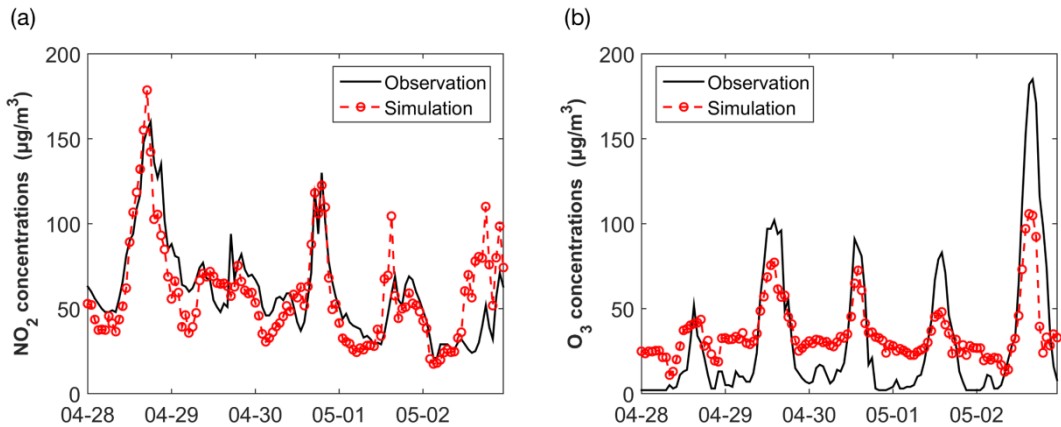

**Figure 12. Time series of (a) NO₂ and (b) O₃ during the simulation period. (black solid line: observation; red dashed line: simulation).**

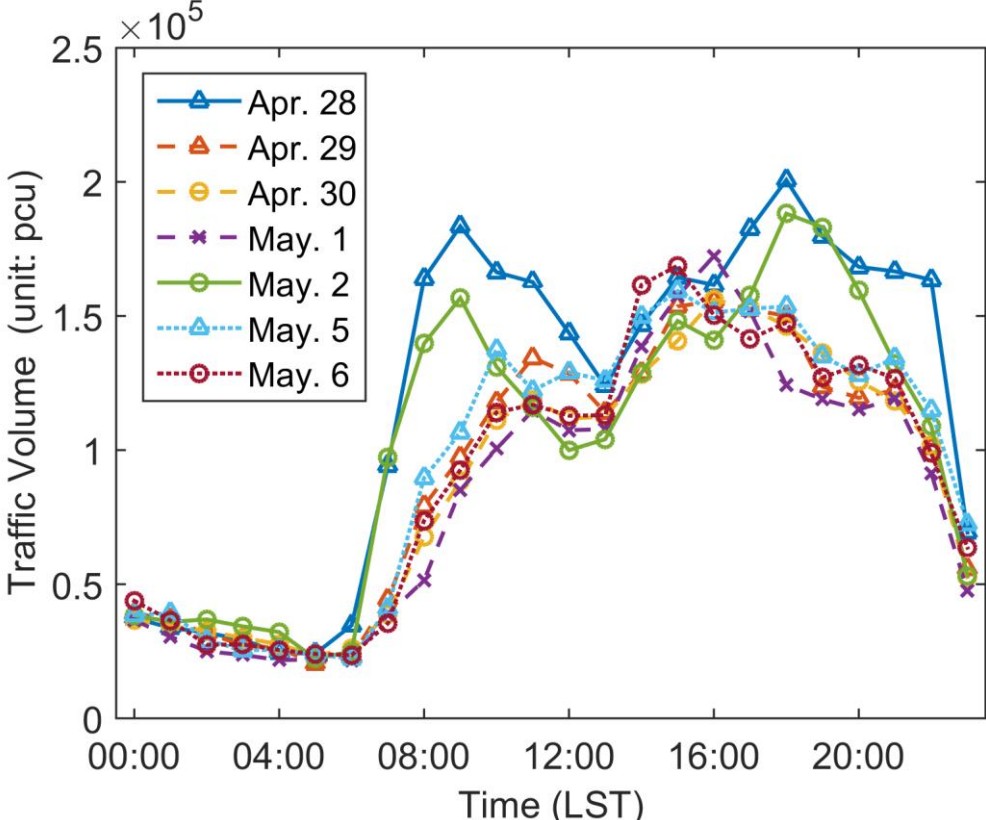

**Figure 13. The diurnal variation of the total traffic volume in the simulation street network (solid line: normal weekday; dashed line: national holiday; dotted line: normal weekend).**

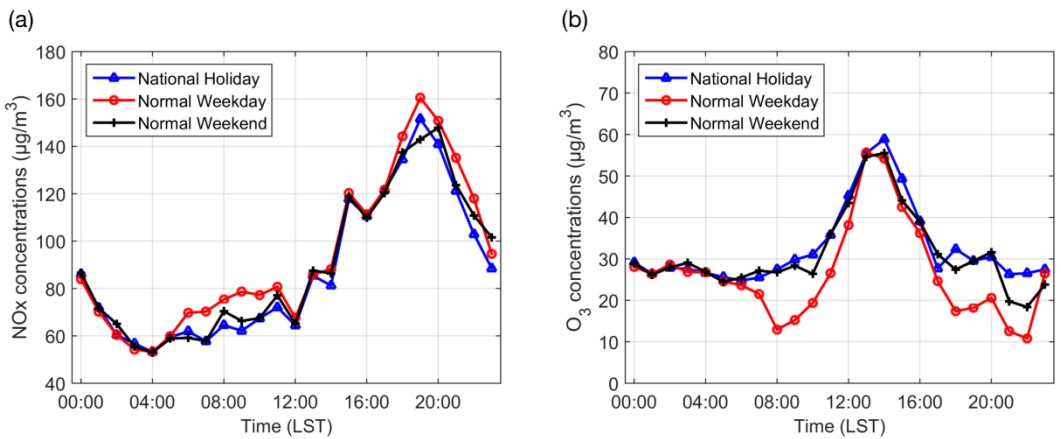

**Figure 14. The (a) NOₓ and (b) O₃ diurnal variation of different sensitivity cases in the simulation street network**