# Peer review of "Development of a real-time on-road emission (ROE v1.0) model for street-scale air quality modeling based on dynamic traffic big data"

_Geoscientific Model Development, 2019_

## Referee Comment (RC1) · Anonymous Referee #1 · 22 May 2019

Review Comments on Luo et al. (GMD-2019-74)

Major comments

This paper describes a new emission model (ROE) that is based on the bottom-up approach and the intelligent transport system (ITS) in China and that can generate real-time high-resolution emissions from vehicles in Guang zhou area. It also describes the application of an urban street-network model, MUNICH, in Guangzhou area using the traffic emissions estimated by the ROE. Street-level emission and air quality modeling is an important and hot topic in atmospheric sciences yet technically and computationally challenging. This paper fits the scopes of GMD and EGU and addresses this very

important topics. It represents one of the first applications of the MUNICH model in China. The approaches for the development of ROE and the application of MNICH are technically sound. The results are very interesting and illustrated the skills of MUNICH in simulating surface NOx and O3 concentrations in the Guang zhou area. The paper is well organized. The methodologies and results from ROE and MUNICH are overall well described, although more detailed and in-depth descriptions are expected in several places (see below and specific comments).

Several important revisions are needed to bring this paper up to the quality for publication in GMD.

1. More detailed descriptions on the development of ROM are needed. For example, what are the assumptions used? What are the uncertainties associated with emission factors? What are the limitations of the ROE that warrant future improvement? How is the ROE developed in this work different from that ITS work of Xiong et al., 2010 over Guang Zhou and also by other people over China? What are the innovative features and uniqueness of this work in the context of existing work?

2. How were the urban background concentrations be derived for the MUNICH model? What are the uncertainties/measurement errors associated with the measurements of NOx and O3 concentrations?

3. What are the model evaluation criteria (e.g., threshold values for the statistical metrics) used to judge the model performance? How are those statistics compared with other model evaluation for simulated NOx and O3 concentrations reported in the literature?

4. More in-depth discussions of emission modeling results are needed, e.g., the discussion for Table 2 that compares the three emission datasets. Why are NOx emissions estimated in this work higher than those from the other two? Why are the differences in gaseous emissions larger than those in PM2.5/PM10 emissions among the three inventories? Also, it would be useful to provide a brief description on the basis of

[Figure]

MEIC-2016 and PRD-2015 inventories, which may help understand the differences across the three inventories.

5. More in-depth discussions of air quality modeling results are needed, e.g., discussion for Figure 11, why does the model give larger NOx overpredictions of NOx and O3 underpredictions on May 2? It looks that the model tends to overpredict O3 mixing ratios at night, could you please explain the likely causes for this overprediction? Does this error come from the overestimated urban background O3, or underpredicted NOx titration (as the model tends to underpredict NOx mixing ratios at night) or both? Is it possible to set up some sensitivity simulations to verify/pin-point your speculated causes for the model bias?

6. In the conclusion section, it would be useful to discuss the limitations of this work and future areas of improvement for both ROE and the MUNICH modeling work.

7. The paper contains some grammatical errors, typos, and undefined acronyms. Additional references should be cited in several places. It would benefit from an editorial review by an English-speaker.

In sum, this paper represents an important contribution to the street-level emission and air quality modeling. I would recommend the paper be accepted after the authors revise the paper to fully address the above major comments and the specific comments below.

Specific comments

1. Page 1, line 16, does "Mg/a" mean "Mg/year"? If so, I would suggest to use "Mg/yr". A similar question for "Mg/a" in page 5, line 36.

2. Page 1, line 32, replace "has seen" by "has experienced".

3. Page 1, lines 33-34, "Zheng et al., 2009b" should be cited after "Zheng et al., 2009a"

4. Page 1, lines 35-36, are those percentages concentrations or emissions of CO,

NOx, and HC? Please clarify.

5. Page 1, line 37, "Numerical emission modeling" (rather than "Numerical air quality modeling") is "an effective method to estimate on-road vehicle emissions", please correct this.

6. Page 2, line 13, replace "leads" by "lead"

7. Page 2, line 18, replace "heavy reliance" by "strong dependence"

8. Page 2, line 29, replace "observation" by "observational"

9. Page 3, lines 16-20, how is the ROE developed in this work different from that ITS work of Xiong et al., 2010 over Guang Zhou and also by other people over China? What are the innovative features and uniqueness of this work in the context of existing work?

10. Page 4, please discuss uncertainties associated with emission factors.

11. Page 4, lines 21-27, please explain why the Underwood volume calculation model was selected. This method was developed about 60-year ago, is it still better than more recent methods?

12. Page 4, line 41, replace "includes" by "include"

13. Page 5, line 1, "7:00-22:00" covers not only daytime but also nighttime, please clarify.

14. Page 5, after Section 2.4, it would be useful to discuss any limitation and uncertainties associated with the ROE model. Page 2, lines 41-44 indicated some issues with the ITS methods, are those issues applicable to the ROE developed for Guangzhou area? Also, what specific traffic information and emission factors will be needed if one applies the approaches/modules used in the ROE model to estimate real-time traffic emissions in other cities?

15. Page 5, line 12, which version of WRF was used?

16. Page 5, lines 18-19, why were only 31 main street segments selected?

17. Page 5, line 20, please spell out "WUDAPT".

18. Page 5, line 24, why was "the 28th April 2018 to the 2nd May 2018" selected? This needs to be explained up front, not in a section later.

19. Page 5, lines 26-28, are "boundary conditions" the same as the urban background concentrations needed for MUNICH model simulations? How are the measured NOx and O3 concentrations used to derive the boundary conditions? What are the uncertainties/measurement errors associated with those measurements?

20. Page 5, lines 33-37 and page 6, lines 1-2. Please add some discussions on the comparison of the three emission datasets in Table 2, e.g., why are the NOx emissions estimated in this work higher than those from the other two? Why are the differences in gaseous emissions larger than those in PM2.5/PM10 emissions among the three inventories? Also, it would be useful to provide a brief description on the basis of MEIC-2016 and PRD-2015 inventories which may help understand the differences across the three inventories. Were MEIC-2016 and PRD-2015 based on the top-down or bottom up approaches? Can those differences be related to the limitations associated with the emission modeling methods discussed in page 2?

21. Page 7, lines 7-8, discussion for Figure 11, could you explain why the model gives larger NOx overpredictions of NOx and O3 underpredictions on May 2? It looks that the model tends to overpredict O3 mixing ratios at night, could you please explain the likely causes for this overprediction? Does this error come from the overestimated urban background O3, or underpredicted NOx titration (as the model tends to underpredict NOx mixing ratios at night) or both? Is it possible to set up some sensitivity simulations to verify/pin-point your speculated causes for the model bias?

22. Page 7, lines 7-16, a reference is needed for those selected metrics. What are

the criteria used to judge the model performance to be good? How are those statistics compared with other model evaluation for simulated NOx and O3 concentrations reported in the literature?

23. Page 7, line 11, "the model overestimated values", do "values" mean "observations"? Please replace "a MB" by "an MB".

24. Page 7, line 12, please remove "respective"

25. Page 7, line 13, please add "respectively" after "0.90", replace "values" by "observations", replace "a MB" by "an MB".

26. Page 7, line 36, replace "As Table 5 shows" by "As shown in Table 5"

27. Page 8, line 12, replace "the emission" by "the emissions"

28. Page 8, line 42, replace "observation" by "observational"

29. Page 8, it would be useful to discuss the limitations of this work and future areas of improvement for both ROE and the MUNICH modeling work.

30. Page 12, Table 1, please provide the full name of acronyms such as RRTM, ACM2, UCM in the footnote and references for each module. Please also indicate the version of WRF used in the table title.

31. Page 13, Table 4, "RESE" should be "RMSE". Please add a footnote to define all acronyms such as OBS, SIM, etc. to make the table self-explainable.

32. Page 13, Tables 5-6, it is not necessary to include "%" in all numbers in those tables. Suggest to delete "%" from all numbers in the tables and add "percentage" before "differences" in the title of the tables.

---

## Short Comment (SC1) · 24 May 2019

I am writing as an executive editor of GMD to highlight issues with the code availability section which needs to be remedied in the revised manuscript.

Thank you for providing the code associated with your manuscript. There are a few issues which are outstanding.

[Figure]

**Code is only on GitHub**

Thank you for providing a reference to the full code and data used in the experiments presented in your manuscript. There are two problems with providing this data via GitHub. The first is that a reader cannot identify the exact version of the code that was used in the paper (for example, you may fix bugs or add features in the future). The second issue is that projects sometimes change the revision control system they use, or the hosting (the project might move to GitLab, for example). The solution to both of these issues is to provide a reference to a persistent archive of the exact version of the code that was used in the manuscript. This reference can, and should, be in addition to the GitHub link, so that a user can also always access the most recent version of the code.

Since your original code is hosted on GitHub, the easiest way to produce a persistent archive of a precise version is to use GitHub's Zenodo integration. For more details, see: https://guides.github.com/activities/citable-code/.

Please ensure that the revised version of your manuscript contains a reference to a persistent, public archive of the exact version of the code used to produce it.

**It is not clear whether the data and scripts are complete**

Your code should include all of the data required to produce the results in the manuscript, including any data processing and plotting scripts. It is not completely clear to me from reading the code whether this is the case here. Please ensure that all of this data is present, and that the documentation in the repository is sufficiently clear that a user will know how to use the code and data provided to reproduce the results in the manuscript.

---

## Referee Comment (RC2) · Anonymous Referee #2 · 25 May 2019

I - General comments

This manuscript aims to present a new modelling approach to represent traffic emissions at street-scale and a first application of this approach within the Guangzhou city.

The relevance of this new methodology is evaluated through the comparison to observations of simulation results of a street-scale air quality model for NO2 and O3.

Finally a first application aiming to quantify the impact of traffic volume change on street-scale photochemistry is analysed.

Its a topic of scientific interest and within the scope of Geoscientific Model Development. The presentation of the work is globally very clearly organised, but sometimes too limited to a simple description. A more in depth presentation of some crucial points (see below) would be very useful to improve this manuscript.

II - Detailed Comments

p2 line 39: The authors mention "The real-time traffic data from the road network could be the most precise input data for on-road emission inventories and could significantly improve the spatial and temporal resolution of the inventories."

I believe this is the central point of the work presented. The targeted question is to know if "real" traffic data can help to improve the quality of modelled emissions and then the quality of modelled concentrations.

Of course the first step is to be able to use such data. The work presented shows it is technically possible. The second step is to show that it allows to get reasonable emissions and then reasonable concentrations. The manuscript provides some elements for this second step.

However what is missing in this work, from my point of view, is the demonstration of the interest of the proposed methodology in comparison to previously existing methods. It could have been relevant to compare a simulation with the emissions derived from the new methodology to a simulation with emissions derived from time-averaged data and applying hourly, daily and monthly factors as often applied within Top-Down approaches. Similarly the comparison to spatially-averaged data (at a chosen grid cell scale and per type of ways) would be of interest.

p3 line 4: Strictly speaking it is not the case for all air pollutants. This sentence could be rewritten avoiding this useless generalisation.

p3 line 26: This sentence should be rewritten to clarify what is available currently (it is always possible to develop a model to extend its functionalities).

p4 line 15 and section 2.2: More details on the emission factors building methodolo-

Interactive
comment

gies would be useful to appreciate their relevance in a near real-time / "instantaneous" framework. Does their temporal representativeness is fully consistent with the fine temporal description of the traffic data? If not, what are the expected impact on the results?

p5 line 33 : The table 2 only shows the global results without any analysis. I guess a comprehensive comparison of the three inventories is beyond the scope of the current paper, however some general considerations and analysis concerning the discrepancies between the three database appears mandatory for this manuscript.

p5 line 35 and followings : The numbers provided in tables should not be recalled in the body text.

p7 line 14-15 : From section 3 I understand the "boundary conditions" are considered to feed the MUNICH runs. It implies that others sources than on-road emissions are implicitly considered.

p8 line 35 : One of the traditional aim of models is to be used for prospective (long term forecast) studies. Could the authors provide some hints on how their methodology could be extended too perform such study?

---

## Author Comment (AC1) · 3 Jul 2019

Response to reviewer

We are very thankful to reviewer for his/her constructive criticisms and valuable comments, which were of great help in improving the quality of the manuscript. We have revised the manuscript accordingly and our detailed responses are shown below. The responses are also shown in the supplement to this comment.

Referee 1

Response to major comments

1. More detailed descriptions on the development of ROM are needed. For example, what are the assumptions used? What are the uncertainties associated with emission factors? What are the limitations of the ROE that warrant future improvement? How is the ROE developed in this work different from that ITS work of Xiong et al., 2010 over Guangzhou and also by other people over China? What are the innovative features and uniqueness of this work in the context of existing work?

Reply: Thanks for pointing out this. The very basic assumption of the ROE model was that calculating the street-level emissions from every on-road vehicle on the street segment by using the bottom-up method. The ROE model collected the traffic information from ITS for obtaining the traffic volume and vehicle fleet information from street network. After obtaining the number of each vehicle category, the ROE calculated the emissions of every vehicle category and sum it up for obtaining the total emission for each street segment. In this study, due to the vehicle fleet information (the emission standard information, vehicle category information and fuel type information) is not available from the data source we used, we used a uniform percentage, which is the average value of the Guangzhou city for each segment. This could be update if the street-level fleet information was available in the future. In order to make the model description clearer, we added section 2.1 "model overview" in page 3 line 24-30 and added more details in section 2.2.

The uncertainty analysis of emission factors had added in supplementary materials section S2. As shown in Figure S1, the uncertainty range of LDV was the largest for CO, HC and NOx. Besides, the HDT has the largest uncertainty for PM2.5 and PM10, whether it was petrol or diesel. However, more comprehensive emissions factor measurements should be done to analyze the uncertainty and improve the accuracy of the emission factors in the future.

In supplementary materials section S2, "In order to estimate the uncertainty of emission factors, some results from previous studies had been collected and compared with the emission factors which were applied in this study. As shown in Figure S1, the

uncertainties for each pollutant of petrol vehicles were much larger than the diesel vehicles. Overall, the uncertainties of LDVs for CO, HC, and NOx were largest than other vehicle categories, whether it was petrol- or diesel-fueled. There were maximum 8.9 and 9.8 times higher for CO emission factors, 13.5 and 21.9 for HC, and 10.5 and 2.0 times for NOx than that of the emission factors applied in this study. And for PM2.5 and PM10, the HDTs had the largest uncertainty range, which were maximum 11.9 and 11.3 times higher for petrol HDT, and 3.5 and 16.1 times for higher diesel HDT, compared with the emission factors used in this study, respectively. However, it should be indicated more comprehensive emissions factor measurements should be done to analyze the uncertainty and improve the accuracy of the emission factors in the future."

The limitations of the ROE model had discussed in the section 5 Discussion and conclusions part. In page 10, line 20-23, "It is worth noting that the ROE model is highly dependent on the ITS traffic data. For economically underdeveloped cities, this aspect may pose a barrier against the use of the ROE model. In addition, China is promoting the CHINA VI emission standards for on-road vehicles. The ROE model only considers Pre-CHINA I to CHINA V currently. Thus, the model will be updated in the near future to include the CHINA VI emission standards.". Moreover, another limitation is that in page 10, line 1-3, "due to the lack of street-level vehicle fleet information, this study applied a city-level average uniform percentage for every street segment. This may increase the uncertainty of the inventory, but this aspect could be improved upon provided additional data become available in the future."

In page 3, line 17-18, the ITS work of Xiong et al. (2010) over Guangzhou aimed to establish a monitoring system which could obtain the traffic conditions automatically by using the traffic cameras, while the ROE model aimed to establish the high-resolution on-road emission inventories by using the traffic data from ITSs. The ROE model could be applied in other cities if the traffic data are available. If the official data from on-road cameras cannot be obtained, the ROE users can still able to use the data from amap.com (also called Gaode map), which is the same data source as this study used.

The current version of crawler module of ROE was designed for obtaining the data from Gaode map. In page 4, line 31-32, "The Gaode map traffic data are quite extensive as it covered over 40 cities in China so far (with most of them being China's major cities)." This could help the users apply the ROE model in other cities of China. We also had added some detail information about the data source in section 2.4.

The most innovative features and uniqueness of this work is that we used the real-time traffic data to establish the high-resolution emission inventories by bottom-up method, which could obtain the hourly or minutely on-road from some certain street segment. Besides, in page 4, line 32-37, "Based on the GPS and mobile network information, details on vehicle speed and location are collected from the map user's devices while using the map navigation on the road. This aspect saves a considerable amount of human labor and material resources with regard to traffic condition observations. These data are updated in real time and can be used through an open-access application programming interface (API), which remove the barrier of obtaining data. As the data can be updated in real time, the emission data can also be refreshed in real time.".

2. How were the urban background concentrations be derived for the MUNICH model? What are the uncertainties/measurement errors associated with the measurements of NOx and O3 concentrations?

Reply: Thanks for your constructive comments. In page 6, line 18-21, "The observational data from TYX were used for modeling evaluation because TYX locates inside the simulation area which could be compared with the model results. In addition, YJ is located near but not within the simulation street network. The observational data from YJ could be used as the background concentration data for the modeling.".

In page 6, line 12-17, "For modeling evaluation and background concentrations, the observational concentration data for NO2 and O3 were obtained from the Guangzhou environmental monitoring site network. NO2 concentrations were measured with a chemiluminescence instrument (Model 42i, Thermo Scientific) and O3 was measured

by a UV photometric analyzer (Model 49i, Thermo Scientific). The minimum detection limit (3S/N) of the analyzer was 0.4 ppbV (approximately 0.8 $\mu$g/m3) for NO2 and 1.0 ppbV (approximately 2.0 $\mu$g/m3) for O3. The total measurement uncertainty of these two instruments was estimated to be approximately 5% (Zhang et al., 2014)."

3. What are the model evaluation criteria (e.g., threshold values for the statistical metrics) used to judge the model performance? How are those statistics compared with other model evaluation for simulated NOx and O3 concentrations reported in the literature?

Reply: Thank you for providing these important points. We had added the recommended values from Ministry of Ecology and Environment of the People's Republic of China technical guide in section 4.2.1. In page 8, line 23-29, "The NMB, NME, and CORR values of NO2 and O3 in this study were within the recommended ranges in the MEP Technical Guide for Air Quality Model Selection (MEP, 2012). These recommended values were -40% < NMB < 50%, NME < 80% and R2 > 0.3 for NO2, and -15% < NMB < 15%, NME < 35%, and R2 > 0.4 for O3. Additionally, the values obtained in this study fell within the range of those obtained by other modeling studies in Guangzhou; the NMB, NME and RMSE values for simulated urban NO2 in Guangzhou were -27.5% to -6%, 29.2% to 53.0% and 16 to 37.3, respectively, and the corresponding values for O3 were and -21.2% to 20.0%, 38.2% to 98%, 9.4 to 40.1 (Che et al., 2011; Fan et al., 2015; Wang et al., 2016). "

4. More in-depth discussions of emission modeling results are needed, e.g., the discussion for Table 2 that compares the three emission datasets. Why are NOx emissions estimated in this work higher than those from the other two? Why are the differences in gaseous emissions larger than those in PM2.5/PM10 emissions among the three inventories? Also, it would be useful to provide a brief description on the basis of MEIC-2016 and PRD-2015 inventories, which may help understand the differences across the three inventories.

Reply: Thank you for your helpful suggestion. We had added some details about the MEIC and PRD inventories. In page 6, line 29-34, "These two emission inventories used the top-down method to establish on-road emission inventories. Unlike the bottom-up method used in this study, these two inventories first calculated the total emissions based on the VKT data of vehicle categories. In the MEIC inventory, the total number of vehicles was obtained from the relationship between total vehicle ownership and economic development (Zheng et al., 2014), while the PRD inventory acquired information on the number of vehicles from the city-level statistics Yearbook. Then, the spatial distribution of these two inventories was established based on the road network density."

According to the uncertainty analysis of emission factors, the uncertainty of PM2.5 and PM10 is much smaller than the gaseous emissions, leading the large difference of gaseous emissions.

As for NOx emissions, we thought that the higher NOx estimate could be due to our updated LPG bus emission factor based on the local study (Zhang et al., 2013). The NOx emission factor of an LPG-fueled bus is 1.7 times that of a diesel-fueled bus. This maybe one of the reasons leading the higher NOx estimate. From figure 9, the results showed that the NOx emission distribution of bus in urban and suburban area was 20.5% and 10.8%.

We had added this content in page 6, line 38 to page 7, line 4, "the difference of PM2.5 and PM10 amount was smaller than other gaseous emissions among different inventories. This was because that the uncertainty of particulate matter emission factors was lower than the corresponding values of the other emissions, which led to the large difference for the gaseous emissions and the smaller differences for PM2.5 and PM10. For NOx emissions, however, this study showed a higher NOx estimate than that in the other two inventories. One of the reasons for the higher NOx estimate may be the application of the updated LPG bus emission factors in this study. Based on a previous local emission factor study, the NOx emission factor of an LPG-fueled bus is 1.7 times

that of a diesel-fueled bus in Guangzhou (Zhang et al., 2013). The results in Figure 8 show that the NOx emissions distribution attributable to buses in urban and suburban areas were 20.5% and 10.8% of the total NOx, respectively, showing that the LPG-fueled buses may be responsible for higher NOx estimates in this study compared to those in the other two inventories."

5. More in-depth discussions of air quality modeling results are needed, e.g., discussion for Figure 11, why does the model give larger NOx overpredictions of NOx and O3 underpredictions on May 2? It looks that the model tends to overpredict O3 mixing ratios at night, could you please explain the likely causes for this overprediction? Does this error come from the overestimated urban background O3, or underpredicted NOx titration (as the model tends to underpredict NOx mixing ratios at night) or both? Is it possible to set up some sensitivity simulations to verify/pin-point your speculated causes for the model bias?

Reply: Thanks for bringing up this meaningful question. Several sensitivity cases had been further carried out to figure out what affected the simulation results in supplementary materials section S3. We had compared the observational and background concentrations of NO2 and O3 and analyze the results from the model sensitivity cases (Figure S2 to S5 in supplementary materials). We thought that NO2 overprediction during the daytime was caused by the overestimation of background concentrations. Due to the only consideration of on-road emission and overprediction of NO2, the VOCs-to-NOx emission ratio was underestimated. Meanwhile, the simulated street network was in the VOC-sensitive regime. Thus, the O3 concentrations were underpredicted during the daytime, especially on May 2nd. For the nighttime O3 overprediction, both overestimated background concentrations and underestimation of NO titration were the reasons for higher predicted nighttime O3. As shown by the sensitivity case results shown (Figure S2 to S5 in supplementary materials), the underestimated NO titration had also led to a lower simulated NO2 concentrations at night.

In supplementary materials section S3, "In order to figure out what affected the simulation results, some modeling sensitivity cases had been carried out in this study. As the diurnal variations shown in Figure S2, NO2 concentrations were underestimated, while the O3 were overestimated at night. However, the daytime model predictions were opposite to the nighttime prediction with NO2 overprediction and O3 underprediction. Combined with the observation and background concentrations data in Figure S3, the daytime NO2 and nighttime O3 overprediction maybe be caused by the overestimation of background concentrations. To evaluate the impact of background concentrations on the simulation results, the no-emission sensitivity case was carried out for enhancing the background effect. This case was run without any emissions and results were shown in the Figure S4. For NO2, daytime concentrations were still overestimated compared with the observational data, especially in May 2nd, with maximum 23.7% daytime overestimation during the simulation. Meanwhile, nighttime O3 were also overpredicted in this case, showing the nighttime O3 overestimation was mostly due to the overestimation of background O3 concertation. In contrast, since only vehicle emissions were considered, the underestimation of daytime O3 should relate to the lack of other sections of emission in the simulation street network. Besides, another case was carried out to evaluate the nighttime NOx titration. Since NO2 were underestimated at night but overestimated during daytime, NO2 titration was not underpredicted and probably overpredicted at night. The overestimation of nighttime should be due to the underestimation of NO concentration. Thus, the double-background-NO case was carried out with double background NO concentrations to evaluate the NO titration. As the background NO concertation increased, nighttime NO2 had increased which offset the underestimation concentration in base case. Meanwhile, O3 concentrations had decreased due to the enhancement of NO titration. These results had shown that on the one hand, the overestimation of background O3 concentration could lead to the O3 overprediction at night. On the other hand, the underestimated NO titration was also a reason for the nighttime overprediction of O3 concentrations."

We had summarized up the results from those sensitivity case. The conclusions were shown in page 8, line 16-19, "Generally, the overestimated background concentrations

of NO2 and O3 caused the reason for the overprediction of daytime NO2 and night-time O3 concentrations, respectively. Also, the underestimated NO titration was the other main reason for overprediction of O3 and underprediction of NO2 concentrations at night. Due to the only consideration of on-road emission in the simulation street network, daytime O3 concentrations were underpredicted in the results.".

6. In the conclusion section, it would be useful to discuss the limitations of this work and future areas of improvement for both ROE and the MUNICH modeling work.

Reply: Thank you for the reminder. The limitations of the ROE modeling work are listed as below: In page 10, line 1-3, "due to the lack of street-level vehicle fleet information, this study applied a city-level average uniform percentage for every street segment. This may increase the uncertainty of the inventory, but this aspect could be improved upon provided additional data become available in the future." In page 10, line 21-23, "In addition, China is promoting the CHINA VI emission standards for on-road vehicles. The ROE model only considers Pre-CHINA I to CHINA V currently. Thus, the model will be updated in the near future to include the CHINA VI emission standards.".

As for MUNICH modeling work, in page 10, line 13-18, "In this study, only 31 main street segments were selected to study the impact of a holiday on air quality in a certain urban area of Guangzhou. Additional investigations are required to understand the variations in street-level air quality in urban or suburban area of a megacity. The results of the ROE model showed that the suburban town centers of Guangzhou served as emission hotspots. These areas had relatively higher emissions than the other suburban areas and less stringent control policies than the urban area, which suffers from more serious air quality problems."

7. The paper contains some grammatical errors, typos, and undefined acronyms. Additional references should be cited in several places. It would benefit from an editorial review by an English-speaker.

Reply: We had carefully rechecked the language problem and added more references

for making the paper clearer.

Response to specific comments

1. Page 1, line 16, does "Mg/a" mean "Mg/year"? If so, I would suggest to use "Mg/yr". A similar question for "Mg/a" in page 5, line 36.

Reply: Agreed. We had revised it in page1, line 16 and page 9 line 34.

2. Page 1, line 32, replace "has seen" by "has experienced".

Reply: Agreed. We had revised it in page 1, line 32.

3. Page 1, lines 33-34, "Zheng et al., 2009b" should be cited after "Zheng et al., 2009a"

Reply: Agreed. We had revised it in page 1, line 33-34.

4. Page 1, lines 35-36, are those percentages concentrations or emissions of CO, NOx, and HC? Please clarify.

Reply: These percentages were the emissions of CO, NOx and HC. We had revised it in page 1, line 35-36.

5. Page 1, line 37, "Numerical emission modeling" (rather than "Numerical air quality modeling") is "an effective method to estimate on-road vehicle emissions", please correct this.

Reply: In here, what we want to express is that the on-road emission inventory can be used as input data for the numerical air quality models which were applied to estimate the impact of on-road emissions on the air quality. We though it's more suitable to use "numerical air quality modeling" here instead of "Numerical emission modeling". To avoid the misunderstanding, we had rewritten the sentence by "Reliable on-road emission inventories can be used as input data for the numerical air quality models which were applied to estimate the impact of on-road emissions on the urban air quality." in page 1, line 38-39.

6. Page 2, line 13, replace "leads" by "lead"

Reply: Agreed. We had revised it in page2, line 13

7. Page 2, line 18, replace "heavy reliance" by "strong dependence"

Reply: Agreed. We had revised it in page2, line 17.

8. Page 2, line 29, replace "observation" by "observational"

Reply: Agreed. We had revised it in page2, line 29.

9. Page 3, lines 16-20, how is the ROE developed in this work different from that ITS work of Xiong et al., 2010 over Guang Zhou and also by other people over China? What are the innovative features and uniqueness of this work in the context of existing work?

Reply: Thanks for bringing up this meaningful question. The ITS work of Xiong et al. (2010) over Guangzhou aimed to establish a monitoring system which could obtain the traffic conditions automatically by using the traffic cameras, while the ROE model aimed to establish the high-resolution on-road emission inventories by using the traffic data from ITSs. The ROE model could be applied in other cities if the traffic data are available. If the official data from on-road cameras cannot be obtained, the ROE users can still able to use the data from amap.com (also called Gaode map), which is the same data source as this study used. The current version of crawler module of ROE was designed for obtaining the data from Gaode map. In page 4, line 31-32, "The Gaode map traffic data are quite extensive as it covered over 40 cities in China so far (with most of them being China's major cities)." This could help the users apply the ROE model in other cities of China. We also had added some detail information about the data source in section 2.4.

The most innovative features and uniqueness of this work is that we used the real-time traffic data to establish the high-resolution emission inventories by bottom-up method, which could obtain the hourly or minutely on-road from some certain street segment.

Besides, in page 4, line 32-37, "Based on the GPS and mobile network information, details on vehicle speed and location are collected from the map user's devices while using the map navigation on the road. This aspect saves a considerable amount of human labor and material resources with regard to traffic condition observations. These data are updated in real time and can be used through an open-access application programming interface (API), which remove the barrier of obtaining data. As the data can be updated in real time, the emission data can also be refreshed in real time.".

10. Page 4, please discuss uncertainties associated with emission factors.

Reply: Thanks for pointing out this. We had discussed the uncertainties of emission factors in the supplementary materials section S2. As shown in Figure S1, the uncertainty range of LDV was the largest for CO, HC and NOx. Besides, the HDT has the largest uncertainty for PM2.5 and PM10, whether it was petrol or diesel. However, more comprehensive emissions factor measurements should be done to analyze the uncertainty and improve the accuracy of the emission factors in the future.

In supplementary materials section S2, "In order to estimate the uncertainty of emission factors, some results from previous studies had been collected and compared with the emission factors which were applied in this study. As shown in Figure S1, the uncertainties for each pollutant of petrol vehicles were much larger than the diesel vehicles. Overall, the uncertainties of LDVs for CO, HC, and NOx were largest than other vehicle categories, whether it was petrol- or diesel-fueled. There were maximum 8.9 and 9.8 times higher for CO emission factors, 13.5 and 21.9 for HC, and 10.5 and 2.0 times for NOx than that of the emission factors applied in this study. And for PM2.5 and PM10, the HDTs had the largest uncertainty range, which were maximum 11.9 and 11.3 times higher for petrol HDT, and 3.5 and 16.1 times for higher diesel HDT, compared with the emission factors used in this study, respectively. However, it should be indicated more comprehensive emissions factor measurements should be done to analyze the uncertainty and improve the accuracy of the emission factors in the future"

11. Page 4, lines 21-27, please explain why the Underwood volume calculation model was selected. This method was developed about 60-year ago, is it still better than more recent methods?

Reply: In the work of Jing et al. (2016), they tested several speed-flow models and found the Underwood model had best goodness of fit among these models in China megacity. Thus, we applied this speed-flow model in our work. In page 4, line 41-43, we had clarified that "In this study, the Underwood volume calculation model was used to retrieve the information on volume because of its history of successful application in China (Jing et al., 2016)."

12. Page 4, line 41, replace "includes" by "include"

Reply: Agreed. We had rewritten the sentence by "The main traffic control policies in urban areas are as follows:" in page 5, line 17-18.

13. Page 5, line 1, "7:00-22:00" covers not only daytime but also nighttime, please clarify.

Reply: Agreed. We had revised it in page 5, line 20.

14. Page 5, after Section 2.4, it would be useful to discuss any limitation and uncertainties associated with the ROE model. Page 2, lines 41-44 indicated some issues with the ITS methods, are those issues applicable to the ROE developed for Guangzhou area? Also, what specific traffic information and emission factors will be needed if one applies the approaches/modules used in the ROE model to estimate real-time traffic emissions in other cities?

Reply: The limitations of the ROE model had discussed in the section 5 Discussion and conclusions part. In page 10, line 20-23, "It is worth noting that the ROE model is highly dependent on the ITS traffic data. For economically underdeveloped cities, this aspect may pose a barrier against the use of the ROE model. In addition, China is promoting the CHINA VI emission standards for on-road vehicles. The ROE model only

considers Pre-CHINA I to CHINA V currently. Thus, the model will be updated in the near future to include the CHINA VI emission standards.". Moreover, another limitation is that in page 10, line 1-3, "due to the lack of street-level vehicle fleet information, this study applied a city-level average uniform percentage for every street segment. This may increase the uncertainty of the inventory, but this aspect could be improved upon provided additional data become available in the future."

The ROE model applied the bottom-up method to establish the on-road emission inventory. To apply the ROE model in other cities, in the best case, as it introduced in section 2.1, "First, the ROE model collects the real-time traffic information to obtain the traffic volume for each street segment from the ITS. Then, according to the vehicle fleet information, the ROE model calculates the number of vehicles for each vehicle category on each street segment (if available, these data could be obtained from the ITS and need not be calculated by model). Thereafter, the ROE model calculates the emissions for street segments based on the vehicle fleet information, traffic conditions, and environmental conditions.", the street-level traffic volume for each vehicle category (e.g., the number of each vehicle type, the percentage of emission standard and fuel type for each vehicle type), traffic speed of street-segments and street information (e.g., street length, street width) are needed in the model. As shown in section 2.4, If the traffic volume data are unavailable, users could refer to the method of this study for obtaining the traffic speed data from the Gaode map and traffic volume data by the speed-flow model. The Gaode map traffic data are quite extensive as it covered over 40 cities in China so far (with most of them being China's major cities). Then, users could use the city-level average uniform percentage of emission standard and fuel type to obtain the volume of each vehicle category. For the emission factors, as shown in section 2.3, users could apply the recommended value from the MEP guidebook, which are also listed in the supplementary materials. Those emission factors are national-wide and can be used in other city of China. User could also update the emission factors once they have their own data.

This would ensure the normal use of the ROE model in different cities.

15. Page 5, line 12, which version of WRF was used?

Reply: The WRF version we used in this study was 3.7.1. We had revised it in page5, line 34.

16. Page 5, lines 18-19, why were only 31 main street segments selected?

Reply: These 31 main street segments all locate in the Center Business District (CBD) which has significant diurnal traffic variation compared with other district in urban area. The reason that we selected these 31 main street segments was clarified in page 5, line 41 to page 6, line 3, "The simulation area comprised 31 main street segments selected to simulate the variation in pollutant concentrations, because continuous traffic data existed for these street segments during the simulation period which were representative within the street network."

17. Page 5, line 20, please spell out "WUDAPT".

Reply: We had revised it in page 6, line 4

18. Page 5, line 24, why was "the 28th April 2018 to the 2nd May 2018" selected? This needs to be explained up front, not in a section later.

Reply: Agreed. We had revised it in page 6, line 8-11, "The simulation period of the study spanned from the April 28th, 2018 to the May 2nd, 2018. There was a significate traffic volume change between holidays and non-holidays. This simulation period covered holidays and non-holidays, which was helpful to investigate the impact of traffic volume variations on air quality."

19. Page 5, lines 26-28, are "boundary conditions" the same as the urban background concentrations needed for MUNICH model simulations? How are the measured NOx and O3 concentrations used to derive the boundary conditions? What are the uncertainties/measurement errors associated with those measurements?

[Figure]

Reply: The boundary conditions were the same as background concentrations which was needed for MUNICH model. We had changed the "boundary conditions" to "background concentrations" for easier understanding in page 6, line 12 and line 21.

The background concentrations were from a monitoring site (YangJi site, YJ) outside but near the simulation street network. The observational data from this site could provide the background concentrations for MUNICH model. We had clarified it in page 6, line 20-21, "In addition, YJ is located near but not within the simulation street network. The observational data from YJ could be used as the background concentration data for the modeling.".

The uncertainties of the measurements were declared in page 6, line 12-17, "For modeling evaluation and background concentrations, the observational concentration data for NO2 and O3 were obtained from the Guangzhou environmental monitoring site network. NO2 concentrations were measured with a chemiluminescence instrument (Model 42i, Thermo Scientific) and O3 was measured by a UV photometric analyzer (Model 49i, Thermo Scientific). The minimum detection limit (3S/N) of the analyzer was 0.4 ppbV (approximately 0.8 $\mu$g/m3) for NO2 and 1.0 ppbV (approximately 2.0 $\mu$g/m3) for O3. The total measurement uncertainty of these two instruments was estimated to be approximately 5% (Zhang et al., 2014)."

20. Page 5, lines 33-37 and page 6, lines 1-2. Please add some discussions on the comparison of the three emission datasets in Table 2, e.g., why are the NOx emissions estimated in this work higher than those from the other two? Why are the differences in gaseous emissions larger than those in PM2.5/PM10 emissions among the three inventories? Also, it would be useful to provide a brief description on the basis of MEIC-2016 and PRD-2015 inventories which may help understand the differences across the three inventories. Were MEIC-2016 and PRD-2015 based on the top-down or bottom up approaches? Can those differences be related to the limitations associated with the emission modeling methods discussed in page 2?

[Figure]

Reply: Thank you for your helpful suggestion. We had added some details about the MEIC and PRD inventories. In page 6, line 29-34, "These two emission inventories used the top-down method to establish on-road emission inventories. Unlike the bottom-up method used in this study, these two inventories first calculated the total emissions based on the VKT data of vehicle categories. In the MEIC inventory, the total number of vehicles was obtained from the relationship between total vehicle ownership and economic development (Zheng et al., 2014), while the PRD inventory acquired information on the number of vehicles from the city-level statistics Yearbook. Then, the spatial distribution of these two inventories was established based on the road network density."

According to the uncertainty analysis of emission factors, the uncertainty of PM2.5 and PM10 is much smaller than the gaseous emissions, leading the large difference of gaseous emissions.

As for NOx emissions, we thought that the higher NOx estimate could be due to our updated LPG bus emission factor based on the local study (Zhang et al., 2013). The NOx emission factor of an LPG-fueled bus is 1.7 times that of a diesel-fueled bus. This maybe one of the reasons leading the higher NOx estimate. From figure 9, the results showed that the NOx emission distribution of bus in urban and suburban area was 20.5% and 10.8%.

We had added this content in page 6, line 38 to page 7, line 4, "the difference of PM2.5 and PM10 amount was smaller than other gaseous emissions among different inventories. This was because that the uncertainty of particulate matter emission factors was lower than the corresponding values of the other emissions, which led to the large difference for the gaseous emissions and the smaller differences for PM2.5 and PM10. For NOx emissions, however, this study showed a higher NOx estimate than that in the other two inventories. One of the reasons for the higher NOx estimate may be the application of the updated LPG bus emission factors in this study. Based on a previous local emission factor study, the NOx emission factor of an LPG-fueled bus is 1.7 times

that of a diesel-fueled bus in Guangzhou (Zhang et al., 2013). The results in Figure 8 show that the NOx emissions distribution attributable to buses in urban and suburban areas were 20.5% and 10.8% of the total NOx, respectively, showing that the LPG-fueled buses may be responsible for higher NOx estimates in this study compared to those in the other two inventories."

21. Page 7, lines 7-8, discussion for Figure 11, could you explain why the model gives larger NOx overpredictions of NOx and O3 underpredictions on May 2? It looks that the model tends to overpredict O3 mixing ratios at night, could you please explain the likely causes for this overprediction? Does this error come from the overestimated urban background O3, or underpredicted NOx titration (as the model tends to underpredict NOx mixing ratios at night) or both? Is it possible to set up some sensitivity simulations to verify/pin-point your speculated causes for the model bias?

Reply: Thanks for bringing up this meaningful question. Several sensitivity cases had been further carried out to figure out what affected the simulation results in supplementary materials section S3. We had compared the observational and background concentrations of NO2 and O3 and analyze the results from the model sensitivity cases (Figure S2 to S5 in supplementary materials). We thought that NO2 overprediction during the daytime was caused by the overestimation of background concentrations. Due to the only consideration of on-road emission and overprediction of NO2, the VOCs-to-NOx emission ratio was underestimated. Meanwhile, the simulated street network was in the VOC-sensitive regime. Thus, the O3 concentrations were underpredicted during the daytime, especially on May 2nd. For the nighttime O3 overprediction, both overestimated background concentrations and underestimation of NO titration were the reasons for higher predicted nighttime O3. As shown by the sensitivity case results shown (Figure S2 to S5 in supplementary materials), the underestimated NO titration had also led to a lower simulated NO2 concentrations at night.

In supplementary materials section S3, "In order to figure out what affected the simulation results, some modeling sensitivity cases had been carried out in this study. As the

diurnal variations shown in Figure S2, NO2 concentrations were underestimated, while the O3 were overestimated at night. However, the daytime model predictions were opposite to the nighttime prediction with NO2 overprediction and O3 underprediction. Combined with the observation and background concentrations data in Figure S3, the daytime NO2 and nighttime O3 overprediction maybe be caused by the overestimation of background concentrations. To evaluate the impact of background concentrations on the simulation results, the no-emission sensitivity case was carried out for enhancing the background effect. This case was run without any emissions and results were shown in the Figure S4. For NO2, daytime concentrations were still overestimated compared with the observational data, especially in May 2nd, with maximum 23.7% daytime overestimation during the simulation. Meanwhile, nighttime O3 were also overpredicted in this case, showing the nighttime O3 overestimation was mostly due to the overestimation of background O3 concertation. In contrast, since only vehicle emissions were considered, the underestimation of daytime O3 should relate to the lack of other sections of emission in the simulation street network. Besides, another case was carried out to evaluate the nighttime NOx titration. Since NO2 were underestimated at night but overestimated during daytime, NO2 titration was not underpredicted and probably overpredicted at night. The overestimation of nighttime should be due to the underestimation of NO concentration. Thus, the double-background-NO case was carried out with double background NO concentrations to evaluate the NO titration. As the background NO concertation increased, nighttime NO2 had increased which offset the underestimation concentration in base case. Meanwhile, O3 concentrations had decreased due to the enhancement of NO titration. These results had shown that on the one hand, the overestimation of background O3 concentration could lead to the O3 overprediction at night. On the other hand, the underestimated NO titration was also a reason for the nighttime overprediction of O3 concentrations."

We had summarized up the results from those sensitivity case. The conclusions were shown in page 8, line 16-19, "Generally, the overestimated background concentrations of NO2 and O3 caused the reason for the overprediction of daytime NO2 and nighttime O3 concentrations, respectively. Also, the underestimated NO titration was the other main reason for overprediction of O3 and underprediction of NO2 concentrations at night. Due to the only consideration of on-road emission in the simulation street network, daytime O3 concentrations were underpredicted in the results.".

22. Page 7, lines 7-16, a reference is needed for those selected metrics. What are the criteria used to judge the model performance to be good? How are those statistics compared with other model evaluation for simulated NOx and O3 concentrations reported in the literature?

Reply: Thank you for providing these important points. We had added the recommended values from Ministry of Ecology and Environment of the People's Republic of China technical guide in section 4.2.1. In page 8, line 23-29, "The NMB, NME, and CORR values of NO2 and O3 in this study were within the recommended ranges in the MEP Technical Guide for Air Quality Model Selection (MEP, 2012). These recommended values were -40% < NMB < 50%, NME < 80% and R2 > 0.3 for NO2, and -15% < NMB < 15%, NME < 35%, and R2 > 0.4 for O3. Additionally, the values obtained in this study fell within the range of those obtained by other modeling studies in Guangzhou; the NMB, NME and RMSE values for simulated urban NO2 in Guangzhou were -27.5% to -6%, 29.2% to 53.0% and 16 to 37.3, respectively, and the corresponding values for O3 were and -21.2% to 20.0%, 38.2% to 98%, 9.4 to 40.1 (Che et al., 2011; Fan et al., 2015; Wang et al., 2016). "

23. Page 7, line 11, "the model overestimated values", do "values" mean "observations"? Please replace "a MB" by "an MB".

Reply: We had rewritten this paragraph deleted this sentence.

24. Page 7, line 12, please remove "respective"

Reply: Agreed. We had rewritten this paragraph deleted this sentence.

25. Page 7, line 13, please add "respectively" after "0.90", replace "values" by "observations", replace "a MB" by "an MB".

Reply: We had rewritten this paragraph deleted this sentence.

26. Page 7, line 36, replace "As Table 5 shows" by "As shown in Table 5"

Reply: Agreed. We had revised it in page 9, line 8.

27. Page 8, line 12, replace "the emission" by "the emissions"

Reply: Agreed. We had revised it in page 9, line 27.

28. Page 8, line 42, replace "observation" by "observational"

Reply: Agreed. We had revised it in page 10, line 35.

29. Page 8, it would be useful to discuss the limitations of this work and future areas of improvement for both ROE and the MUNICH modeling work.

Reply: Thank you for the reminder. The limitations of the ROE modeling work are listed as below: In page 10, line 1-3, "due to the lack of street-level vehicle fleet information, this study applied a city-level average uniform percentage for every street segment. This may increase the uncertainty of the inventory, but this aspect could be improved upon provided additional data become available in the future." In page 10, line 21-23, "In addition, China is promoting the CHINA VI emission standards for on-road vehicles. The ROE model only considers Pre-CHINA I to CHINA V currently. Thus, the model will be updated in the near future to include the CHINA VI emission standards.".

As for MUNICH modeling work, in page 10, line 13-18, "In this study, only 31 main street segments were selected to study the impact of a holiday on air quality in a certain urban area of Guangzhou. Additional investigations are required to understand the variations in street-level air quality in urban or suburban area of a megacity. The results of the ROE model showed that the suburban town centers of Guangzhou served as emission hotspots. These areas had relatively higher emissions than the other suburban areas and less stringent control policies than the urban area, which suffers from more serious

air quality problems."

30. Page 12, Table 1, please provide the full name of acronyms such as RRTM, ACM2, UCM in the footnote and references for each module. Please also indicate the version of WRF used in the table title.

Reply: Agreed. We had revised it page 15, Table 1.

31. Page 13, Table 4, "RESE" should be "RMSE". Please add a footnote to define all acronyms such as OBS, SIM, etc. to make the table self-explainable.

Reply: Agreed. We had revised it in page 16, Table 4.

32. Page 13, Tables 5-6, it is not necessary to include "%" in all numbers in those tables. Suggest to delete "%" from all numbers in the tables and add "percentage" before "differences" in the title of the tables.

Reply: Agreed. We had revised it in page 16, Table 5-6.

Reference Che, W., Zheng, J., Wang, S., Zhong, L. and Lau, A.: Assessment of motor vehicle emission control policies using Model-3/CMAQ model for the Pearl River Delta region, China, Atmos. Environ., 45(9), 1740–1751, doi:10.1016/j.atmosenv.2010.12.050, 2011. Fan, Q., Lan, J., Liu, Y., Wang, X., Chan, P., Hong, Y., Feng, Y., Liu, Y., Zeng, Y. and Liang, G.: Process analysis of regional aerosol pollution during spring in the Pearl River Delta region, China, Atmos. Environ., 122(January 2013), 829–838, doi:10.1016/j.atmosenv.2015.09.013, 2015. Jing, B., Wu, L., Mao, H., Gong, S., He, J., Zou, C., Song, G., Li, X. and Wu, Z.: Development of a vehicle emission inventory with high temporal-spatial resolution based on NRT traffic data and its impact on air pollution in Beijing - Part 1: Development and evaluation of vehicle emission inventory, Atmos. Chem. Phys., 16(5), 3161–3170, doi:10.5194/acp-16-3161-2016, 2016. Wang, N., Lyu, X. P., Deng, X. J., Guo, H., Deng, T., Li, Y., Yin, C. Q., Li, F. and Wang, S. Q.: Assessment of regional air quality resulting from emission control in the Pearl River Delta region, southern China, Sci. Total Environ., 573(11),

1554–1565, doi:10.1016/j.scitotenv.2016.09.013, 2016. Xiong, G., Wang, K., Zhu, F., Cheng, C., An, X. and Xie, Z.: Parallel traffic management for the 2010 Asian Games, IEEE Intell. Syst., 25(3), 81–85, doi:10.1109/MIS.2010.87, 2010. Zhang, G., Mu, Y., Liu, J., Zhang, C., Zhang, Y., Zhang, Y. and Zhang, H.: Seasonal and diurnal variations of atmospheric peroxyacetyl nitrate, peroxypropionyl nitrate, and carbon tetrachloride in Beijing, J. Environ. Sci., 26(1), 65–74, doi:10.1016/S1001-0742(13)60382-4, 2014. Zhang, S., Wu, Y., Liu, H., Wu, X., Zhou, Y., Yao, Z., Fu, L., He, K. and Hao, J.: Historical evaluation of vehicle emission control in Guangzhou based on a multi-year emission inventory, Atmos. Environ., 76, 32–42, doi:10.1016/j.atmosenv.2012.11.047, 2013. Zheng, B., Huo, H., Zhang, Q., Yao, Z. L., Wang, X. T., Yang, X. F., Liu, H. and He, K. B.: High-resolution mapping of vehicle emissions in China in 2008, Atmos. Chem. Phys., 14(18), 9787–9805, doi:10.5194/acp-14-9787-2014, 2014.

Please also note the supplement to this comment:
https://www.geosci-model-dev-discuss.net/gmd-2019-74/gmd-2019-74-AC1-supplement.pdf
* * *

---

## Author Comment (AC2) · 3 Jul 2019

Response to reviewer

We are very thankful to reviewer for his/her constructive criticisms and valuable comments, which were of great help in improving the quality of the manuscript. We have revised the manuscript accordingly and our detailed responses are shown below. The responses are also shown in the supplement to this comment.

Referee 2

1. p2 line 39: The authors mention "The real-time traffic data from the road network

could be the most precise input data for on-road emission inventories and could significantly improve the spatial and temporal resolution of the inventories." I believe this is the central point of the work presented. The targeted question is to know if "real" traffic data can help to improve the quality of modelled emissions and then the quality of modelled concentrations. Of course the first step is to be able to use such data. The work presented shows it is technically possible. The second step is to show that it allows to get reasonable emissions and then reasonable concentrations. The manuscript provides some elements for this second step. However what is missing in this work, from my point of view, is the demonstration of the interest of the proposed methodology in comparison to previously existing methods. It could have been relevant to compare a simulation with the emissions derived from the new methodology to a simulation with emissions derived from time-averaged data and applying hourly, daily and monthly factors as often applied within Top-Down approaches. Similarly the comparison to spatially-averaged data (at a chosen grid cell scale and per type of ways) would be of interest.

Reply: Thank you for your helpful suggestion. We had known that this is important to compare our results with other existing emission inventories as many details as we can, no matter those inventories were establish by top-down or bottom-up method. However, due to the lack of data of the other inventories, some comparisons are unable to achieve in this study. Even so, we had tried out best to add more details about the comparison between our results and others. In section 4.1.2, we had compared the spatial distribution with other two inventories. In page 7, line 26-32, "Moreover, the spatial distributions of these three emission inventories were compared in this study. Figure 10 shows the distribution of CO from the three different inventories. In urban areas, the results of both MEIC-2016 and PRD-2015 showed the urban areas as emission hotspots. However, the results from the ROE model were much lower for such areas. This may be due to the fact that the ROE model considers the traffic control policies, while the other two inventories do not. In suburban town centers, especially in the eastern and southern parts of Guangzhou, all three inventories showed the same

results, namely that these areas were large contributors of on-road emissions. Notably, highways and arterial roads also contributed high emissions in all three inventories."

"However, it should be noted that this comparison was only preliminary; the spatial resolutions of three inventories are inconsistent. Moreover, due to the lack of temporal information in the other two emission inventories, a comparison of the temporals difference could not be conducted. Future studies should focus on improving the accuracy of such comparisons." (In page 9, line 37-40)

2. p3 line 4: Strictly speaking it is not the case for all air pollutants. This sentence could be rewritten avoiding this useless generalisation.

Reply: Thanks for pointing out this. We had revised the sentence by "Many studies have successfully applied the regional-scale CTMs to investigate the impact of on-road vehicles on the air quality in urban areas in the regional scale (∼100 km)." in page 3, line 4-5.

3. p3 line 26: This sentence should be rewritten to clarify what is available currently (it is always possible to develop a model to extend its functionalities).

Reply: Thanks for your constructive comment. We had revised by adding a sentence "The current version of the ROE model includes the crawler module for the from amap.com (also called the Gaode map) application (Figure 2), a widely adopted map application in China (additional details are provided in section 2.4.)." to clarify the available data source in the current version of model in page 3, line 38-40.

4. p4 line 15 and section 2.2: More details on the emission factors building methodologies would be useful to appreciate their relevance in a near real-time / "instantaneous" framework. Does their temporal representativeness is fully consistent with the fine temporal description of the traffic data? If not, what are the expected impact on the results?

Reply: Thanks for pointing out this. We had shown the emission factors and their

correction factors in supplementary materials section S1. We had considered the traffic condition correction factors (Table S9 and Table S10) when we calculated the on-road emissions. The emission factors are different under different traffic conditions and the classification of these traffic conditions are based on the real-time traffic data. These correction factors were clarified in page 4, line 22-25, "The correction factors involving environmental conditions (e.g., temperature, relative humidity, and altitude) and traffic conditions obtained from the technical guide were considered in the study. They are listed in Tables S4–S10 in the supplementary materials. These correction factors were applied to reduce the effects of uncertainties associated with the emission factors."

5. p5 line 33: The table 2 only shows the global results without any analysis. I guess a comprehensive comparison of the three inventories is beyond the scope of the current paper, however some general considerations and analysis concerning the discrepancies between the three database appears mandatory for this manuscript.

Reply: Thank you for your helpful suggestion. We had added some details about the MEIC and PRD inventories. In page 6, line 29-34, "These two emission inventories used the top-down method to establish on-road emission inventories. Unlike the bottom-up method used in this study, these two inventories first calculated the total emissions based on the VKT data of vehicle categories. In the MEIC inventory, the total number of vehicles was obtained from the relationship between total vehicle ownership and economic development (Zheng et al., 2014), while the PRD inventory acquired information on the number of vehicles from the city-level statistics Yearbook. Then, the spatial distribution of these two inventories was established based on the road network density."

According to the uncertainty analysis of emission factors, the uncertainty of PM2.5 and PM10 is much smaller than the gaseous emissions, leading the large difference of gaseous emissions.

As for NOx emissions, we thought that the higher NOx estimate could be due to our

updated LPG bus emission factor based on the local study (Zhang et al., 2013). The NOx emission factor of an LPG-fueled bus is 1.7 times that of a diesel-fueled bus. This maybe one of the reasons leading the higher NOx estimate. From figure 9, the results showed that the NOx emission distribution of bus in urban and suburban area was 20.5% and 10.8%.

We had added this content in page 6, line 38 to page 7, line 4, "the difference of PM2.5 and PM10 amount was smaller than other gaseous emissions among different inventories. This was because that the uncertainty of particulate matter emission factors was lower than the corresponding values of the other emissions, which led to the large difference for the gaseous emissions and the smaller differences for PM2.5 and PM10. For NOx emissions, however, this study showed a higher NOx estimate than that in the other two inventories. One of the reasons for the higher NOx estimate may be the application of the updated LPG bus emission factors in this study. Based on a previous local emission factor study, the NOx emission factor of an LPG-fueled bus is 1.7 times that of a diesel-fueled bus in Guangzhou (Zhang et al., 2013). The results in Figure 8 show that the NOx emissions distribution attributable to buses in urban and suburban areas were 20.5% and 10.8% of the total NOx, respectively, showing that the LPG-fueled buses may be responsible for higher NOx estimates in this study compared to those in the other two inventories."

6. p5 line 35 and followings: The numbers provided in tables should not be recalled in the body text.

Reply: Agreed. We had deleted it in section 4.1.1.

7. p7 line 14-15: From section 3 I understand the "boundary conditions" are considered to feed the MUNICH runs. It implies that other sources than on-road emissions are implicitly considered.

Reply: Yes. The "boundary conditions" represented the "background concentrations" from outside the simulation street network. To make it understand more easily, we had

changed the "boundary conditions" to "background concentrations" in page 6, line and line 21.

8. p8 line 35: One of the traditional aim of models is to be used for prospective (long term forecast) studies. Could the authors provide some hints on how their methodology could be extended too perform such study?

Reply: This is a good point. In discussions and conclusions part, we had discussed the possibility of applying the street-level air quality model in forecasting the variation of pollutants. In page 10, line 24-28, "Recently studies had shown that traffic forecasting models are effective within cities (Min et al., 2009; Cortez et al., 2012; Vlahogianni et al., 2014). These models allow one to obtain predicted traffic-based on-road emissions. Combined with the meteorological forecasting systems and regional air quality forecasting systems, which provide the meteorological and background concentration predictions, respectively, street-level air quality models could be used for street-level air quality forecasting as well.".

Reference Cortez, P., Rio, M., Rocha, M. and Sousa, P.: Multi-scale Internet traffic forecasting using neural networks and time series methods, Expert Syst., 29(2), 143–155, doi:10.1111/j.1468-0394.2010.00568.x, 2012 Min, X., Hu, J., Chen, Q., Zhang, T. and Zhang, Y.: Short-term traffic flow forecasting of urban network based on dynamic STARIMA model, IEEE Conf. Intell. Transp. Syst. Proceedings, ITSC, 100084, 461–466, doi:10.1109/ITSC.2009.5309741, 2009. Vlahogianni, E. I., Karlaftis, M. G. and Golias, J. C.: Short-term traffic forecasting: Where we are and where we're going, Transp. Res. Part C Emerg. Technol., 43, 3–19, doi:10.1016/j.trc.2014.01.005, 2014. Zhang, S., Wu, Y., Liu, H., Wu, X., Zhou, Y., Yao, Z., Fu, L., He, K. and Hao, J.: Historical evaluation of vehicle emission control in Guangzhou based on a multi-year emission inventory, Atmos. Environ., 76, 32–42, doi:10.1016/j.atmosenv.2012.11.047, 2013. Zheng, B., Huo, H., Zhang, Q., Yao, Z. L., Wang, X. T., Yang, X. F., Liu, H. and He, K. B.: High-resolution mapping of vehicle emissions in China in 2008, Atmos. Chem. Phys., 14(18), 9787–9805, doi:10.5194/acp-14-9787-2014, 2014.

Please also note the supplement to this comment:
https://www.geosci-model-dev-discuss.net/gmd-2019-74/gmd-2019-74-AC2-supplement.pdf

---

## Author Comment (AC3) · 3 Jul 2019

**ROE model User Guide**
* * *
ROE

Real-time on-road emission model

This is an on-road emission calculation model

Author: Luolin Wu

Contact: wuluolin@mail2.sysu.edu.cn
* * *
0.   Preparing the Python environment

This model is established and run in Python3.
Internet access is requested for crawler module to obtain the traffic data from ITS.

1.   Crawling data
In this current version, the crawler module is design for Gaode map.
The module folder is '***gaode-crawler***'

**1)   *crawler-traffic-speed.py**

This is the main script of the crawler module. In this script, four options should be modified by users based on the actual situation:

'*path*' is the option for setting the output file path.
'*apikey*' is the option for setting the Gaode map API key which is for accessing the data for Gaode. To obtain the key, please Register an account at https://lbs.amap.com/.
'*citycode*' is the option for setting the city identifier. Please refer the citycode table.
'*target_area*' is the option for setting the crawling area. Four parameters are need for this option. ('*left bottom lon, left bottom lat, right top lon, right ton lat*')

**2)   *LocaDiv.py**

This script is designed for separating the target area into multiple small areas because the Gaode API has a limitation that each request area could not larger than 100km$^2$. Thus, this script could solve this problem if the target area is larger than the limitation. In this script, the small area size could be set in '*lon_len*' and '*lat_len*'.

**3)   *coordTrans.py**

This script is designed for coordinate transformation. The coordinate would be transformed to WGS84.

2.   Processing the traffic data for emission calculation
2.1 Preparing for the base road map
After obtaining the traffic speed from API, the data need to be processed for emission calculation. A road map should be prepared for matching the road information and traffic information because those data from Gaode API do not have the street information (e.g. road type, road length). The street information could be obtained from OpenStreetMap dataset. To match the street information more effective with the traffic information, using GIS software is recommended for this spatial analyst.

This street information should be placed in the folder '***base_info***'.
The file is named by '***base_road_map.csv***'.

2.2 Preparing for the vehicle fleet information

The vehicle fleet information is essential for emission calculation. Vehicle number, fuel type and emission standard information for each type of vehicle should be prepared in the model. In current version of ROE model, the uniform percentages for vehicle type, fuel type and emission standard are used in every street segment.

The vehicle fleet information should be placed in the folder **'base_info'**.

1) Vehicle number for each classification should be placed in file **'vehicle_fleet.csv'**. The content of this file should contain the vehicle type name and their number in two columns with header line. More details could be found in the template files.

2) fuel type percentage should be placed in file **'fuel type.csv'**. The content of this file should contain the vehicle type name and fuel type percentages in $n+1$ columns with header line ($n$ represents the number of fuel type). The percentages of all fuel types are required for each type of the vehicle. More details could be found in the template files.

3) emission standard percentage should be placed in file **'emission_standard_percentage.csv'**. The content of this file should contain the vehicle type name and emission standard percentages in m+1 columns with header line (m represents the number of emission standard). The percentages of all emission standards are required for each type of the vehicle. More details could be found in the template files.

2.3 Preparing for the emission factors

The emission factors in ROE model are from the Ministry of Ecology and Environment of the People's Republic of China (MEP) Technical Guide of Air Pollutant Emission Inventory for On Road Vehicles (MEP, 2014).

The emission factors should be placed in the folder **'base_info'**.

1) Emission factors should be placed in file **'On Road Hot Emission Factor Chinese.csv'**. The content of this file should contain the *vehicle type name, vehicle fuel type, vehicle size segment, emission standard, emission control technology, pollutant type and emission factor* in six columns with header line. More details could be found in the template files.

2) HC evaporation emission factors for petrol vehicle should be placed in file **'evaporation-emission-factor.csv'**. The content of this file should contain the *vehicle type name, vehicle fuel type, vehicle size segment, temperature range, and HC evaporation factor* in five columns with header line. More details could be found in the template files.

2.4 Preparing the Traffic volume data

1) Matching the temporal frequency
   The temporal frequency of traffic data should be processed and match the temporal resolution that the users need.

This script is placed in in the folder *'calculation\script'*.
This script is named by *'speed_mean.py'*.

In this script, the traffic data would be processed in hourly.
Options *start_day* and *end_day* should be modified to set the time period of the script running.
Option *step_in_hour* is the temporal frequency of traffic data (e.g. the *step_in_hour*=12 means temporal frequency of traffic data is every 5-minute)

Option *work_path* is for setting the working directory.
Option *data_path* is for setting the traffic data directory.
Option *output_path* is for setting the output directory for this script.
Option *road_map_file* is the file path of base road map.

2) Calculating the traffic volume
   the traffic volume should be calculated by the traffic speed using the traffic speed-flow model.
   The traffic volume calculation script is placed in the folder *'calculation\script'*
   This script is named by *'flow_calculation.py'*.

   This script uses the Underwood volume calculation model (Underwood, 1961) and model is described by the following equation:

   $$V = k_m u \ln \frac{u_f}{u},$$

   In the flow calculation script, the macroscopic fundamental diagram parameters of k and $u_f$ could be modified if users have their own data.

   Options *start_day* and *end_day* should be modified to set the time period of the script running.
   Option *work_path* is for setting the working directory.
   Option *data_path* is for setting the traffic data directory.
   Option *output_path* is for setting the output directory for this script.
   Option *road_map_file* is the file path of base road map.

3. Calculating the emission data
After preparing the traffic data and vehicle fleet information, users can the emission calculation script to establish the emission data.

The emission calculation script is placed in the folder *'calculation/script/'*
This script is named by *'emission_cakculation-china.py'*.

In this script,
Option *work_path* is for setting the working directory.
Option *base_info_path* is for setting the base info files directory.
Option *road_map_file* is the file path of base road map.
Option *data_path* is for setting the traffic data directory.

Option *output_path* is for setting the output directory for this script.

Options *start_day* and *end_day* should be modified to set the time period of the script running.

It's should be noted that,

The *vehicle_fleet_flag_segment, vehicle_fleet_flag, fuel_type_flag, emission_standard_flag* and *the species_flag* must match the information in those base_info files.

The emission calculation script is depended on another script that considering the correction factors of the emission factor. Users could modify these correction factor based on the actual situation. The correction factors could be referred in the MEP Technical Guide. (MEP, 2014).

This script is placed in the folder *'calculation/script/'*

This script is named by *'ef_equation_china.py'*.

4. Post-processing the emission data

After calculating the emissions, ROE provides some post-processing scripts for outputting the emission information and modifying the format for air quality models.

The post-processing scripts are placed in the folder *'post-script'*

1) *Modify_emis.py*

   This script is designed to modify the emission format from ROE to MUNICH model.

   In this script,

   Option *work_path* is for setting the working directory.

   Option *data_path* is for setting the input emission data directory.

   Option *output_path* is for setting the output directory for this script.

2) *Road_type_flow.py*

   This script is designed to output the traffic flow data for each type of the road.

   In this script,

   Option *work_path* is for setting the working directory.

   Option *data_path* is for setting the input traffic flow data directory.

   Option *output_path* is for setting the output directory for this script.

3) *Road_type_speed.py*

   This script is designed to output the traffic speed data for each type of the road.

   In this script,

   Option *work_path* is for setting the working directory.

   Option *data_path* is for setting the input traffic speed data directory.

   Option *output_path* is for setting the output directory for this script.

Folder tree:

```
ROE
├─── base_info
│    ├─── base_road_map.csv
│    ├─── emission_standard_percentage.csv
│    ├─── evaporation-emission-factor.csv
│    ├─── fuel_type_percentage.csv
│    ├─── On Road Hot Emission Factor Chinese.csv
│    └─── vehicle_fleet.csv
├─── calculation
│    ├─── emission_hour
│    ├─── flow_hour
│    │    └─── plot_flow.m
│    ├─── script
│    │    ├─── ef_equation_china.py
│    │    ├─── emis_calculation-china.py
│    │    ├─── flow_calculation.py
│    │    └─── speed_mean.py
│    └─── speed_hour
│         └─── plot_speed.m
├─── gaode-crawler
│    ├─── coordTrans.py
│    ├─── crawler-traffic-speed.py
│    ├─── gaode-city-code
│    │    └─── AMap_adcode_citycode.xlsx
│    ├─── LocaDiv.py
│    └─── output
├─── post-script
│    ├─── modify_emis.py
│    ├─── road_type_flow.py
│    └─── road_type_speed.py
└─── user_guide-v1.1.pdf
```

Reference

MEP: Technical Guide of Air Pollutant Emission Inventory for On Road Vehicles (Trial), Beijing, China., 2014.

Underwood, R. T.: Speed, volume, and density relationship: quality and theory of traffic flow, Yale Bur. Highw. Traffic, 141–188, 1961.

---

## Author Response (AR1)

**Response to reviewers**

**Title: Development of a real-time on-road emission (ROE v1.0) model for street-scale air quality modeling based on dynamic traffic big data (GMD-2019-74)**

We are very thankful to both reviewers for their constructive criticisms and valuable comments, which are of great help in improving the quality of the manuscript. We have revised the manuscript accordingly and our detailed responses are shown below.

**Reply to Reviewers:**

**Referee 1**

Response to major comments

1.    More detailed descriptions on the development of ROM are needed. For example, what are the assumptions used? What are the uncertainties associated with emission factors? What are the limitations of the ROE that warrant future improvement? How is the ROE developed in this work different from that ITS work of Xiong et al., 2010 over Guangzhou and also by other people over China? What are the innovative features and uniqueness of this work in the context of existing work?

Reply:
Thanks for pointing out this. The very basic assumption of the ROE model was that calculating the street-level emissions from every on-road vehicle on the street segment by using the bottom-up method. The ROE model collected the traffic information from ITS for obtaining the traffic volume and vehicle fleet information from street network. After obtaining the number of each vehicle category, the ROE calculated the emissions of every vehicle category and sum it up for obtaining the total emission for each street segment. In this study, due to the vehicle fleet information (the emission standard information, vehicle category information and fuel type information) is not available from the data source we used, we used a uniform percentage, which is the average value of the Guangzhou city for each segment. This could be update if the street-level fleet information was available in the future. In order to make the model description clearer, we added section 2.1 *"model overview"* in page 3 line 24-30 and added more details in section 2.2.

The uncertainty analysis of emission factors had added in supplementary materials section S2. As shown in Figure S1, the uncertainty range of LDV was the largest for CO, HC and $NO_x$. Besides, the HDT has the largest uncertainty for $PM_{2.5}$ and $PM_{10}$, whether it was petrol or diesel. However, more comprehensive emissions factor measurements should be done to analyze the uncertainty and improve the accuracy of the emission factors in the future.

In supplementary materials section S2, *"In order to estimate the uncertainty of emission factors, some results from previous studies had been collected and compared with the emission factors which were applied in this study. As shown in Figure S1, the uncertainties for each pollutant of petrol*

*vehicles were much larger than the diesel vehicles. Overall, the uncertainties of LDVs for CO, HC, and NO$_x$ were largest than other vehicle categories, whether it was petrol- or diesel-fueled. There were maximum 8.9 and 9.8 times higher for CO emission factors, 13.5 and 21.9 for HC, and 10.5 and 2.0 times for NO$_x$ than that of the emission factors applied in this study. And for PM$_{2.5}$ and PM$_{10}$, the HDTs had the largest uncertainty range, which were maximum 11.9 and 11.3 times higher for petrol HDT, and 3.5 and 16.1 times for higher diesel HDT, compared with the emission factors used in this study, respectively. However, it should be indicated more comprehensive emissions factor measurements should be done to analyze the uncertainty and improve the accuracy of the emission factors in the future."*

The limitations of the ROE model had discussed in the section 5 Discussion and conclusions part. In page 10, line 20-23, *"It is worth noting that the ROE model is highly dependent on the ITS traffic data. For economically underdeveloped cities, this aspect may pose a barrier against the use of the ROE model. In addition, China is promoting the CHINA VI emission standards for on-road vehicles. The ROE model only considers Pre-CHINA I to CHINA V currently. Thus, the model will be updated in the near future to include the CHINA VI emission standards."*. Moreover, another limitation is that in page 10, line 1-3, *"due to the lack of street-level vehicle fleet information, this study applied a city-level average uniform percentage for every street segment. This may increase the uncertainty of the inventory, but this aspect could be improved upon provided additional data become available in the future."*

In page 3, line 17-18, the ITS work of Xiong et al. (2010) over Guangzhou aimed to establish a monitoring system which could obtain the traffic conditions automatically by using the traffic cameras, while the ROE model aimed to establish the high-resolution on-road emission inventories by using the traffic data from ITSs. The ROE model could be applied in other cities if the traffic data are available. If the official data from on-road cameras cannot be obtained, the ROE users can still able to use the data from amap.com (also called Gaode map), which is the same data source as this study used. The current version of crawler module of ROE was designed for obtaining the data from Gaode map. In page 4, line 31-32, *"The Gaode map traffic data are quite extensive as it covered over 40 cities in China so far (with most of them being China's major cities)."* This could help the users apply the ROE model in other cities of China. We also had added some detail information about the data source in section 2.4.

The most innovative features and uniqueness of this work is that we used the real-time traffic data to establish the high-resolution emission inventories by bottom-up method, which could obtain the hourly or minutely on-road from some certain street segment. Besides, in page 4, line 32-37, *"Based on the GPS and mobile network information, details on vehicle speed and location are collected from the map user's devices while using the map navigation on the road. This aspect saves a considerable amount of human labor and material resources with regard to traffic condition observations. These data are updated in real time and can be used through an open-access application programming interface (API), which remove the barrier of obtaining data. As the data can be updated in real time, the emission data can also be refreshed in real time."*.

[Figure]

**Figure S1.** Fleet-average emission factors of (a) CO, (b) HC, (c) NO$_x$, (d) PM$_{2.5}$, and (e) PM$_{10}$ (the value of the histogram represented the fleet-average emission factors in the study, the range represented the emission factors from other literatures).

2.  How were the urban background concentrations be derived for the MUNICH model? What are the uncertainties/measurement errors associated with the measurements of NOx and O3 concentrations?

Reply:
Thanks for your constructive comments. In page 6, line 18-21, "*The observational data from TYX were used for modeling evaluation because TYX locates inside the simulation area which could be compared with the model results. In addition, YJ is located near but not within the simulation street network. The observational data from YJ could be used as the background concentration data for the modeling.*".

In page 6, line 12-17, "*For modeling evaluation and background concentrations, the observational concentration data for NO$_2$ and O$_3$ were obtained from the Guangzhou environmental monitoring site network. NO$_2$ concentrations were measured with a chemiluminescence instrument (Model 42i, Thermo Scientific) and O$_3$ was measured by a UV photometric analyzer (Model 49i, Thermo*

*Scientific). The minimum detection limit (3S/N) of the analyzer was 0.4 ppbV (approximately 0.8 μg/m³) for NO₂ and 1.0 ppbV (approximately 2.0 μg/m³) for O₃. The total measurement uncertainty of these two instruments was estimated to be approximately 5% (Zhang et al., 2014).*"

3.    What are the model evaluation criteria (e.g., threshold values for the statistical metrics) used to judge the model performance? How are those statistics compared with other model evaluation for simulated NOx and O3 concentrations reported in the literature?

Reply:

Thank you for providing these important points. We had added the recommended values from Ministry of Ecology and Environment of the People's Republic of China technical guide in section 4.2.1. In page 8, line 23-29, "*The NMB, NME, and CORR values of NO₂ and O₃ in this study were within the recommended ranges in the MEP Technical Guide for Air Quality Model Selection (MEP, 2012). These recommended values were -40% < NMB < 50%, NME < 80% and $R^2 > 0.3$ for NO₂, and -15% < NMB < 15%, NME < 35%, and $R^2 > 0.4$ for O₃. Additionally, the values obtained in this study fell within the range of those obtained by other modeling studies in Guangzhou; the NMB, NME and RMSE values for simulated urban NO₂ in Guangzhou were -27.5% to -6%, 29.2% to 53.0% and 16 to 37.3, respectively, and the corresponding values for O₃ were and -21.2% to 20.0%, 38.2% to 98%, 9.4 to 40.1 (Che et al., 2011; Fan et al., 2015; Wang et al., 2016).*"

4.    More in-depth discussions of emission modeling results are needed, e.g., the discussion for Table 2 that compares the three emission datasets. Why are NOx emissions estimated in this work higher than those from the other two? Why are the differences in gaseous emissions larger than those in PM2.5/PM10 emissions among the three inventories? Also, it would be useful to provide a brief description on the basis of MEIC-2016 and PRD-2015 inventories, which may help understand the differences across the three inventories.

Reply:

Thank you for your helpful suggestion. We had added some details about the MEIC and PRD inventories. In page 6, line 29-34, "*These two emission inventories used the top-down method to establish on-road emission inventories. Unlike the bottom-up method used in this study, these two inventories first calculated the total emissions based on the VKT data of vehicle categories. In the MEIC inventory, the total number of vehicles was obtained from the relationship between total vehicle ownership and economic development (Zheng et al., 2014), while the PRD inventory acquired information on the number of vehicles from the city-level statistics Yearbook. Then, the spatial distribution of these two inventories was established based on the road network density.*"

According to the uncertainty analysis of emission factors, the uncertainty of PM$_{2.5}$ and PM$_{10}$ is much smaller than the gaseous emissions, leading the large difference of gaseous emissions.

As for NO$_x$ emissions, we thought that the higher NO$_x$ estimate could be due to our updated LPG bus emission factor based on the local study (Zhang et al., 2013). The NO$_x$ emission factor of an LPG-fueled bus is 1.7 times that of a diesel-fueled bus. This maybe one of the reasons leading the

higher NO$_x$ estimate. From figure 9, the results showed that the NO$_x$ emission distribution of bus in urban and suburban area was 20.5% and 10.8%.

We had added this content in page 6, line 38 to page 7, line 4, *"the difference of PM$_{2.5}$ and PM$_{10}$ amount was smaller than other gaseous emissions among different inventories. This was because that the uncertainty of particulate matter emission factors was lower than the corresponding values of the other emissions, which led to the large difference for the gaseous emissions and the smaller differences for PM$_{2.5}$ and PM$_{10}$. For NO$_x$ emissions, however, this study showed a higher NO$_x$ estimate than that in the other two inventories. One of the reasons for the higher NO$_x$ estimate may be the application of the updated LPG bus emission factors in this study. Based on a previous local emission factor study, the NO$_x$ emission factor of an LPG-fueled bus is 1.7 times that of a diesel-fueled bus in Guangzhou (Zhang et al., 2013). The results in Figure 8 show that the NO$_x$ emissions distribution attributable to buses in urban and suburban areas were 20.5% and 10.8% of the total NO$_x$, respectively, showing that the LPG-fueled buses may be responsible for higher NO$_x$ estimates in this study compared to those in the other two inventories."*

5. More in-depth discussions of air quality modeling results are needed, e.g., discussion for Figure 11, why does the model give larger NOx overpredictions of NOx and O3 underpredictions on May 2? It looks that the model tends to overpredict O3 mixing ratios at night, could you please explain the likely causes for this overprediction? Does this error come from the overestimated urban background O3, or underpredicted NOx titration (as the model tends to underpredict NOx mixing ratios at night) or both? Is it possible to set up some sensitivity simulations to verify/pin-point your speculated causes for the model bias?

Reply:
Thanks for bringing up this meaningful question. Several sensitivity cases had been further carried out to figure out what affected the simulation results in supplementary materials section S3. We had compared the observational and background concentrations of NO$_2$ and O$_3$ and analyze the results from the model sensitivity cases (Figure S2 to S5 in supplementary materials). We thought that NO$_2$ overprediction during the daytime was caused by the overestimation of background concentrations. Due to the only consideration of on-road emission and overprediction of NO$_2$, the VOCs-to-NO$_x$ emission ratio was underestimated. Meanwhile, the simulated street network was in the VOC-sensitive regime. Thus, the O$_3$ concentrations were underpredicted during the daytime, especially on May 2$^{nd}$. For the nighttime O$_3$ overprediction, both overestimated background concentrations and underestimation of NO titration were the reasons for higher predicted nighttime O$_3$. As shown by the sensitivity case results shown (Figure S2 to S5 in supplementary materials), the underestimated NO titration had also led to a lower simulated NO$_2$ concentrations at night.

In supplementary materials section S3, *"In order to figure out what affected the simulation results, some modeling sensitivity cases had been carried out in this study. As the diurnal variations shown in Figure S2, NO$_2$ concentrations were underestimated, while the O$_3$ were overestimated at night. However, the daytime model predictions were opposite to the nighttime prediction with NO$_2$ overprediction and O$_3$ underprediction. Combined with the observation and background*

*concentrations data in Figure S3, the daytime $NO_2$ and nighttime $O_3$ overprediction maybe be caused by the overestimation of background concentrations.*

*To evaluate the impact of background concentrations on the simulation results, the no-emission sensitivity case was carried out for enhancing the background effect. This case was run without any emissions and results were shown in the Figure S4. For $NO_2$, daytime concentrations were still overestimated compared with the observational data, especially in May $2^{nd}$, with maximum 23.7% daytime overestimation during the simulation. Meanwhile, nighttime $O_3$ were also overpredicted in this case, showing the nighttime $O_3$ overestimation was mostly due to the overestimation of background $O_3$ concertation. In contrast, since only vehicle emissions were considered, the underestimation of daytime $O_3$ should relate to the lack of other sections of emission in the simulation street network.*

*Besides, another case was carried out to evaluate the nighttime $NO_x$ titration. Since $NO_2$ were underestimated at night but overestimated during daytime, $NO_2$ titration was not underpredicted and probably overpredicted at night. The overestimation of nighttime should be due to the underestimation of NO concentration. Thus, the double-background-NO case was carried out with double background NO concentrations to evaluate the NO titration. As the background NO concertation increased, nighttime $NO_2$ had increased which offset the underestimation concentration in base case. Meanwhile, $O_3$ concentrations had decreased due to the enhancement of NO titration. These results had shown that on the one hand, the overestimation of background $O_3$ concentration could lead to the $O_3$ overprediction at night. On the other hand, the underestimated NO titration was also a reason for the nighttime overprediction of $O_3$ concentrations."*

We had summarized up the results from those sensitivity case. The conclusions were shown in page 8, line 16-19, "*Generally, the overestimated background concentrations of $NO_2$ and $O_3$ caused the reason for the overprediction of daytime $NO_2$ and nighttime $O_3$ concentrations, respectively. Also, the underestimated NO titration was the other main reason for overprediction of $O_3$ and underprediction of $NO_2$ concentrations at night. Due to the only consideration of on-road emission in the simulation street network, daytime $O_3$ concentrations were underpredicted in the results.*".

[Figure]

**Figure S2. Diurnal variations of (a) $NO_2$ and (b) $O_3$ during the simulation period. (black solid line: observation; red dashed line: simulation).**

[Figure]

**Figure S3.** Time series of observational and background concentrations for (a) NO₂ and (b) O₃ during the simulation period (black solid line: observation; red dashed line: background).

[Figure]

**Figure S4.** Time series of (a) NO₂ and (b) O₃ during the simulation period in no-emission sensitivity case. (black solid line: observation; red solid line: background).

[Figure]

**Figure S5.** Time series of (a) NO₂ and (b) O₃ during the simulation period in double-background-NO case. (black solid line: observation; red solid line: background).

6.    In the conclusion section, it would be useful to discuss the limitations of this work and future areas of improvement for both ROE and the MUNICH modeling work.

Reply:
Thank you for the reminder. The limitations of the ROE modeling work are listed as below:
In page 10, line 1-3, "*due to the lack of street-level vehicle fleet information, this study applied a city-level average uniform percentage for every street segment. This may increase the uncertainty of the inventory, but this aspect could be improved upon provided additional data become available in the future.*"
In page 10, line 21-23, "*In addition, China is promoting the CHINA VI emission standards for on-road vehicles. The ROE model only considers Pre-CHINA I to CHINA V currently. Thus, the model will be updated in the near future to include the CHINA VI emission standards.*".

As for MUNICH modeling work, in page 10, line 13-18, "*In this study, only 31 main street segments were selected to study the impact of a holiday on air quality in a certain urban area of Guangzhou. Additional investigations are required to understand the variations in street-level air quality in urban or suburban area of a megacity. The results of the ROE model showed that the suburban town centers of Guangzhou served as emission hotspots. These areas had relatively higher emissions than the other suburban areas and less stringent control policies than the urban area, which suffers from more serious air quality problems.*"

7.    The paper contains some grammatical errors, typos, and undefined acronyms. Additional references should be cited in several places. It would benefit from an editorial review by an English-speaker.

Reply:
We had carefully rechecked the language problem and added more references for making the paper clearer.

Response to specific comments

1.    Page 1, line 16, does "Mg/a" mean "Mg/year"? If so, I would suggest to use "Mg/yr". A similar question for "Mg/a" in page 5, line 36.

Reply:
Agreed. We had revised it in page1, line 16 and page 9 line 34.

2.    Page 1, line 32, replace "has seen" by "has experienced".

Reply:
Agreed. We had revised it in page 1, line 32.

3.  Page 1, lines 33-34, "Zheng et al., 2009b" should be cited after "Zheng et al., 2009a"

Reply:
Agreed. We had revised it in page 1, line 33-34.

4.  Page 1, lines 35-36, are those percentages concentrations or emissions of CO, NOx, and HC? Please clarify.

Reply:
These percentages were the emissions of CO, NO$_x$ and HC. We had revised it in page 1, line 35-36.

5.  Page 1, line 37, "Numerical emission modeling" (rather than "Numerical air quality modeling") is "an effective method to estimate on-road vehicle emissions", please correct this.

Reply:
In here, what we want to express is that the on-road emission inventory can be used as input data for the numerical air quality models which were applied to estimate the impact of on-road emissions on the air quality. We though it's more suitable to use "numerical air quality modeling" here instead of "Numerical emission modeling". To avoid the misunderstanding, we had rewritten the sentence by "*Reliable on-road emission inventories can be used as input data for the numerical air quality models which were applied to estimate the impact of on-road emissions on the urban air quality.*" in page 1, line 38-39.

6.  Page 2, line 13, replace "leads" by "lead"

Reply:
Agreed. We had revised it in page2, line 13

7.  Page 2, line 18, replace "heavy reliance" by "strong dependence"

Reply:
Agreed. We had revised it in page2, line 17.

8.  Page 2, line 29, replace "observation" by "observational"

Reply:
Agreed. We had revised it in page2, line 29.

9.  Page 3, lines 16-20, how is the ROE developed in this work different from that ITS work of Xiong et al., 2010 over Guang Zhou and also by other people over China? What are the innovative features and uniqueness of this work in the context of existing work?

Reply:

Thanks for bringing up this meaningful question. The ITS work of Xiong et al. (2010) over Guangzhou aimed to establish a monitoring system which could obtain the traffic conditions automatically by using the traffic cameras, while the ROE model aimed to establish the high-resolution on-road emission inventories by using the traffic data from ITSs. The ROE model could be applied in other cities if the traffic data are available. If the official data from on-road cameras cannot be obtained, the ROE users can still able to use the data from amap.com (also called Gaode map), which is the same data source as this study used. The current version of crawler module of ROE was designed for obtaining the data from Gaode map. In page 4, line 31-32, "*The Gaode map traffic data are quite extensive as it covered over 40 cities in China so far (with most of them being China's major cities).*" This could help the users apply the ROE model in other cities of China. We also had added some detail information about the data source in section 2.4.

The most innovative features and uniqueness of this work is that we used the real-time traffic data to establish the high-resolution emission inventories by bottom-up method, which could obtain the hourly or minutely on-road from some certain street segment. Besides, in page 4, line 32-37, "*Based on the GPS and mobile network information, details on vehicle speed and location are collected from the map user's devices while using the map navigation on the road. This aspect saves a considerable amount of human labor and material resources with regard to traffic condition observations. These data are updated in real time and can be used through an open-access application programming interface (API), which remove the barrier of obtaining data. As the data can be updated in real time, the emission data can also be refreshed in real time.*".

10. Page 4, please discuss uncertainties associated with emission factors.

Reply:
Thanks for pointing out this. We had discussed the uncertainties of emission factors in the supplementary materials section S2. As shown in Figure S1, the uncertainty range of LDV was the largest for CO, HC and $NO_x$. Besides, the HDT has the largest uncertainty for $PM_{2.5}$ and $PM_{10}$, whether it was petrol or diesel. However, more comprehensive emissions factor measurements should be done to analyze the uncertainty and improve the accuracy of the emission factors in the future.

In supplementary materials section S2, "*In order to estimate the uncertainty of emission factors, some results from previous studies had been collected and compared with the emission factors which were applied in this study. As shown in Figure S1, the uncertainties for each pollutant of petrol vehicles were much larger than the diesel vehicles. Overall, the uncertainties of LDVs for CO, HC, and $NO_x$ were largest than other vehicle categories, whether it was petrol- or diesel-fueled. There were maximum 8.9 and 9.8 times higher for CO emission factors, 13.5 and 21.9 for HC, and 10.5 and 2.0 times for $NO_x$ than that of the emission factors applied in this study. And for $PM_{2.5}$ and $PM_{10}$, the HDTs had the largest uncertainty range, which were maximum 11.9 and 11.3 times higher for petrol HDT, and 3.5 and 16.1 times for higher diesel HDT, compared with the emission factors used in this study, respectively. However, it should be indicated more comprehensive emissions factor measurements should be done to analyze the uncertainty and improve the accuracy of the emission factors in the future*"

11. Page 4, lines 21-27, please explain why the Underwood volume calculation model was selected. This method was developed about 60-year ago, is it still better than more recent methods?

Reply:
In the work of Jing et al. (2016), they tested several speed-flow models and found the Underwood model had best goodness of fit among these models in China megacity. Thus, we applied this speed-flow model in our work. In page 4, line 41-43, we had clarified that "*In this study, the Underwood volume calculation model was used to retrieve the information on volume because of its history of successful application in China (Jing et al., 2016).*"

12. Page 4, line 41, replace "includes" by "include"

Reply:
Agreed. We had rewritten the sentence by "*The main traffic control policies in urban areas are as follows:*" in page 5, line 17-18.

13. Page 5, line 1, "7:00-22:00" covers not only daytime but also nighttime, please clarify.

Reply:
Agreed. We had revised it in page 5, line 20.

14. Page 5, after Section 2.4, it would be useful to discuss any limitation and uncertainties associated with the ROE model. Page 2, lines 41-44 indicated some issues with the ITS methods, are those issues applicable to the ROE developed for Guangzhou area? Also, what specific traffic information and emission factors will be needed if one applies the approaches/modules used in the ROE model to estimate real-time traffic emissions in other cities?

Reply:
The limitations of the ROE model had discussed in the section 5 Discussion and conclusions part. In page 10, line 20-23, "*It is worth noting that the ROE model is highly dependent on the ITS traffic data. For economically underdeveloped cities, this aspect may pose a barrier against the use of the ROE model. In addition, China is promoting the CHINA VI emission standards for on-road vehicles. The ROE model only considers Pre-CHINA I to CHINA V currently. Thus, the model will be updated in the near future to include the CHINA VI emission standards.*". Moreover, another limitation is that in page 10, line 1-3, "*due to the lack of street-level vehicle fleet information, this study applied a city-level average uniform percentage for every street segment. This may increase the uncertainty of the inventory, but this aspect could be improved upon provided additional data become available in the future.*"

The ROE model applied the bottom-up method to establish the on-road emission inventory. To apply the ROE model in other cities, in the best case, as it introduced in section 2.1, "*First, the ROE model collects the real-time traffic information to obtain the traffic volume for each street segment from*

*the ITS. Then, according to the vehicle fleet information, the ROE model calculates the number of vehicles for each vehicle category on each street segment (if available, these data could be obtained from the ITS and need not be calculated by model). Thereafter, the ROE model calculates the emissions for street segments based on the vehicle fleet information, traffic conditions, and environmental conditions.*", the street-level traffic volume for each vehicle category (e.g., the number of each vehicle type, the percentage of emission standard and fuel type for each vehicle type), traffic speed of street-segments and street information (e.g., street length, street width) are needed in the model.

As shown in section 2.4, If the traffic volume data are unavailable, users could refer to the method of this study for obtaining the traffic speed data from the Gaode map and traffic volume data by the speed-flow model. The Gaode map traffic data are quite extensive as it covered over 40 cities in China so far (with most of them being China's major cities). Then, users could use the city-level average uniform percentage of emission standard and fuel type to obtain the volume of each vehicle category.

For the emission factors, as shown in section 2.3, users could apply the recommended value from the MEP guidebook, which are also listed in the supplementary materials. Those emission factors are national-wide and can be used in other city of China. User could also update the emission factors once they have their own data.

This would ensure the normal use of the ROE model in different cities.

15. Page 5, line 12, which version of WRF was used?

Reply:
The WRF version we used in this study was 3.7.1. We had revised it in page5, line 34.

16. Page 5, lines 18-19, why were only 31 main street segments selected?

Reply:
These 31 main street segments all locate in the Center Business District (CBD) which has significant diurnal traffic variation compared with other district in urban area. The reason that we selected these 31 main street segments was clarified in page 5, line 41 to page 6, line 3, "*The simulation area comprised 31 main street segments selected to simulate the variation in pollutant concentrations, because continuous traffic data existed for these street segments during the simulation period which were representative within the street network.*"

17. Page 5, line 20, please spell out "WUDAPT".

Reply:
We had revised it in page 6, line 4

18. Page 5, line 24, why was "the 28th April 2018 to the 2nd May 2018" selected? This needs to be explained up front, not in a section later.

Reply:

Agreed. We had revised it in page 6, line 8-11, "*The simulation period of the study spanned from the April 28th, 2018 to the May 2nd, 2018. There was a significate traffic volume change between holidays and non-holidays. This simulation period covered holidays and non-holidays, which was helpful to investigate the impact of traffic volume variations on air quality.*"

19. Page 5, lines 26-28, are "boundary conditions" the same as the urban background concentrations needed for MUNICH model simulations? How are the measured NOx and O3 concentrations used to derive the boundary conditions? What are the uncertainties/measurement errors associated with those measurements?

Reply:

The boundary conditions were the same as background concentrations which was needed for MUNICH model. We had changed the "*boundary conditions*" to "*background concentrations*" for easier understanding in page 6, line 12 and line 21.

The background concentrations were from a monitoring site (YangJi site, YJ) outside but near the simulation street network. The observational data from this site could provide the background concentrations for MUNICH model. We had clarified it in page 6, line 20-21, "*In addition, YJ is located near but not within the simulation street network. The observational data from YJ could be used as the background concentration data for the modeling.*".

The uncertainties of the measurements were declared in page 6, line 12-17, "*For modeling evaluation and background concentrations, the observational concentration data for NO₂ and O₃ were obtained from the Guangzhou environmental monitoring site network. NO₂ concentrations were measured with a chemiluminescence instrument (Model 42i, Thermo Scientific) and O₃ was measured by a UV photometric analyzer (Model 49i, Thermo Scientific). The minimum detection limit (3S/N) of the analyzer was 0.4 ppbV (approximately 0.8 μg/m³) for NO₂ and 1.0 ppbV (approximately 2.0 μg/m³) for O₃. The total measurement uncertainty of these two instruments was estimated to be approximately 5% (Zhang et al., 2014).*"

20. Page 5, lines 33-37 and page 6, lines 1-2. Please add some discussions on the comparison of the three emission datasets in Table 2, e.g., why are the NOx emissions estimated in this work higher than those from the other two? Why are the differences in gaseous emissions larger than those in PM2.5/PM10 emissions among the three inventories? Also, it would be useful to provide a brief description on the basis of MEIC- 2016 and PRD-2015 inventories which may help understand the differences across the three inventories. Were MEIC-2016 and PRD-2015 based on the top-down or bottom up approaches? Can those differences be related to the limitations associated with the emission modeling methods discussed in page 2?

Reply:

Thank you for your helpful suggestion. We had added some details about the MEIC and PRD inventories. In page 6, line 29-34, "*These two emission inventories used the top-down method to*

*establish on-road emission inventories. Unlike the bottom-up method used in this study, these two inventories first calculated the total emissions based on the VKT data of vehicle categories. In the MEIC inventory, the total number of vehicles was obtained from the relationship between total vehicle ownership and economic development (Zheng et al., 2014), while the PRD inventory acquired information on the number of vehicles from the city-level statistics Yearbook. Then, the spatial distribution of these two inventories was established based on the road network density.*"

According to the uncertainty analysis of emission factors, the uncertainty of $PM_{2.5}$ and $PM_{10}$ is much smaller than the gaseous emissions, leading the large difference of gaseous emissions.

As for $NO_x$ emissions, we thought that the higher $NO_x$ estimate could be due to our updated LPG bus emission factor based on the local study (Zhang et al., 2013). The $NO_x$ emission factor of an LPG-fueled bus is 1.7 times that of a diesel-fueled bus. This maybe one of the reasons leading the higher $NO_x$ estimate. From figure 9, the results showed that the $NO_x$ emission distribution of bus in urban and suburban area was 20.5% and 10.8%.

We had added this content in page 6, line 38 to page 7, line 4, *"the difference of $PM_{2.5}$ and $PM_{10}$ amount was smaller than other gaseous emissions among different inventories. This was because that the uncertainty of particulate matter emission factors was lower than the corresponding values of the other emissions, which led to the large difference for the gaseous emissions and the smaller differences for $PM_{2.5}$ and $PM_{10}$. For $NO_x$ emissions, however, this study showed a higher $NO_x$ estimate than that in the other two inventories. One of the reasons for the higher $NO_x$ estimate may be the application of the updated LPG bus emission factors in this study. Based on a previous local emission factor study, the $NO_x$ emission factor of an LPG-fueled bus is 1.7 times that of a diesel-fueled bus in Guangzhou (Zhang et al., 2013). The results in Figure 8 show that the $NO_x$ emissions distribution attributable to buses in urban and suburban areas were 20.5% and 10.8% of the total $NO_x$, respectively, showing that the LPG-fueled buses may be responsible for higher $NO_x$ estimates in this study compared to those in the other two inventories.*"

21.  Page 7, lines 7-8, discussion for Figure 11, could you explain why the model gives larger NOx overpredictions of NOx and O3 underpredictions on May 2? It looks that the model tends to overpredict O3 mixing ratios at night, could you please explain the likely causes for this overprediction? Does this error come from the overestimated urban background O3, or underpredicted NOx titration (as the model tends to underpredict NOx mixing ratios at night) or both? Is it possible to set up some sensitivity simulations to verify/pin-point your speculated causes for the model bias?

Reply:
Thanks for bringing up this meaningful question. Several sensitivity cases had been further carried out to figure out what affected the simulation results in supplementary materials section S3. We had compared the observational and background concentrations of $NO_2$ and $O_3$ and analyze the results from the model sensitivity cases (Figure S2 to S5 in supplementary materials). We thought that $NO_2$ overprediction during the daytime was caused by the overestimation of background concentrations. Due to the only consideration of on-road emission and overprediction of $NO_2$, the VOCs-to-$NO_x$

emission ratio was underestimated. Meanwhile, the simulated street network was in the VOC-sensitive regime. Thus, the $O_3$ concentrations were underpredicted during the daytime, especially on May 2nd. For the nighttime $O_3$ overprediction, both overestimated background concentrations and underestimation of NO titration were the reasons for higher predicted nighttime $O_3$. As shown by the sensitivity case results shown (Figure S2 to S5 in supplementary materials), the underestimated NO titration had also led to a lower simulated $NO_2$ concentrations at night.

In supplementary materials section S3, "*In order to figure out what affected the simulation results, some modeling sensitivity cases had been carried out in this study. As the diurnal variations shown in Figure S2, $NO_2$ concentrations were underestimated, while the $O_3$ were overestimated at night. However, the daytime model predictions were opposite to the nighttime prediction with $NO_2$ overprediction and $O_3$ underprediction. Combined with the observation and background concentrations data in Figure S3, the daytime $NO_2$ and nighttime $O_3$ overprediction maybe be caused by the overestimation of background concentrations.*

*To evaluate the impact of background concentrations on the simulation results, the no-emission sensitivity case was carried out for enhancing the background effect. This case was run without any emissions and results were shown in the Figure S4. For $NO_2$, daytime concentrations were still overestimated compared with the observational data, especially in May 2nd, with maximum 23.7% daytime overestimation during the simulation. Meanwhile, nighttime $O_3$ were also overpredicted in this case, showing the nighttime $O_3$ overestimation was mostly due to the overestimation of background $O_3$ concertation. In contrast, since only vehicle emissions were considered, the underestimation of daytime $O_3$ should relate to the lack of other sections of emission in the simulation street network.*

*Besides, another case was carried out to evaluate the nighttime $NO_x$ titration. Since $NO_2$ were underestimated at night but overestimated during daytime, $NO_2$ titration was not underpredicted and probably overpredicted at night. The overestimation of nighttime should be due to the underestimation of NO concentration. Thus, the double-background-NO case was carried out with double background NO concentrations to evaluate the NO titration. As the background NO concertation increased, nighttime $NO_2$ had increased which offset the underestimation concentration in base case. Meanwhile, $O_3$ concentrations had decreased due to the enhancement of NO titration. These results had shown that on the one hand, the overestimation of background $O_3$ concentration could lead to the $O_3$ overprediction at night. On the other hand, the underestimated NO titration was also a reason for the nighttime overprediction of $O_3$ concentrations.*"

We had summarized up the results from those sensitivity case. The conclusions were shown in page 8, line 16-19, "*Generally, the overestimated background concentrations of $NO_2$ and $O_3$ caused the reason for the overprediction of daytime $NO_2$ and nighttime $O_3$ concentrations, respectively. Also, the underestimated NO titration was the other main reason for overprediction of $O_3$ and underprediction of $NO_2$ concentrations at night. Due to the only consideration of on-road emission in the simulation street network, daytime $O_3$ concentrations were underpredicted in the results.*".

22. Page 7, lines 7-16, a reference is needed for those selected metrics. What are the criteria used to judge the model performance to be good? How are those statistics compared with other model evaluation for simulated NOx and O3 concentrations reported in the literature?

Reply:

Thank you for providing these important points. We had added the recommended values from Ministry of Ecology and Environment of the People's Republic of China technical guide in section 4.2.1. In page 8, line 23-29, "*The NMB, NME, and CORR values of $NO_2$ and $O_3$ in this study were within the recommended ranges in the MEP Technical Guide for Air Quality Model Selection (MEP, 2012). These recommended values were -40% < NMB < 50%, NME < 80% and $R^2$ > 0.3 for $NO_2$, and -15% < NMB < 15%, NME < 35%, and $R^2$ > 0.4 for $O_3$. Additionally, the values obtained in this study fell within the range of those obtained by other modeling studies in Guangzhou; the NMB, NME and RMSE values for simulated urban $NO_2$ in Guangzhou were -27.5% to -6%, 29.2% to 53.0% and 16 to 37.3, respectively, and the corresponding values for $O_3$ were and -21.2% to 20.0%, 38.2% to 98%, 9.4 to 40.1 (Che et al., 2011; Fan et al., 2015; Wang et al., 2016).* "

23.  Page 7, line 11, "the model overestimated values", do "values" mean "observations"? Please replace "a MB" by "an MB".

Reply:
We had rewritten this paragraph deleted this sentence.

24.  Page 7, line 12, please remove "respective"

Reply:
Agreed. We had rewritten this paragraph deleted this sentence.

25.  Page 7, line 13, please add "respectively" after "0.90", replace "values" by "observations", replace "a MB" by "an MB".

Reply:
We had rewritten this paragraph deleted this sentence.

26.  Page 7, line 36, replace "As Table 5 shows" by "As shown in Table 5"

Reply:
Agreed. We had revised it in page 9, line 8.

27.  Page 8, line 12, replace "the emission" by "the emissions"

Reply:
Agreed. We had revised it in page 9, line 27.

28.  Page 8, line 42, replace "observation" by "observational"

Reply:
Agreed. We had revised it in page 10, line 35.

29. Page 8, it would be useful to discuss the limitations of this work and future areas of improvement for both ROE and the MUNICH modeling work.

Reply:
Thank you for the reminder. The limitations of the ROE modeling work are listed as below:
In page 10, line 1-3, "*due to the lack of street-level vehicle fleet information, this study applied a city-level average uniform percentage for every street segment. This may increase the uncertainty of the inventory, but this aspect could be improved upon provided additional data become available in the future.*"
In page 10, line 21-23, "*In addition, China is promoting the CHINA VI emission standards for on-road vehicles. The ROE model only considers Pre-CHINA I to CHINA V currently. Thus, the model will be updated in the near future to include the CHINA VI emission standards.*".

As for MUNICH modeling work, in page 10, line 13-18, "*In this study, only 31 main street segments were selected to study the impact of a holiday on air quality in a certain urban area of Guangzhou. Additional investigations are required to understand the variations in street-level air quality in urban or suburban area of a megacity. The results of the ROE model showed that the suburban town centers of Guangzhou served as emission hotspots. These areas had relatively higher emissions than the other suburban areas and less stringent control policies than the urban area, which suffers from more serious air quality problems.*"

30. Page 12, Table 1, please provide the full name of acronyms such as RRTM, ACM2, UCM in the footnote and references for each module. Please also indicate the version of WRF used in the table title.

Reply:
Agreed. We had revised it page 15, Table 1.

31. Page 13, Table 4, "RESE" should be "RMSE". Please add a footnote to define all acronyms such as OBS, SIM, etc. to make the table self-explainable.

Reply:
Agreed. We had revised it in page 16, Table 4.

32. Page 13, Tables 5-6, it is not necessary to include "%" in all numbers in those tables. Suggest to delete "%" from all numbers in the tables and add "percentage" before "differences" in the title of the tables.

Reply:
Agreed. We had revised it in page 16, Table 5-6.

**Referee 2**

1.  p2 line 39: The authors mention "The real-time traffic data from the road network could be the most precise input data for on-road emission inventories and could significantly improve the spatial and temporal resolution of the inventories."

I believe this is the central point of the work presented. The targeted question is to know if "real" traffic data can help to improve the quality of modelled emissions and then the quality of modelled concentrations.

Of course the first step is to be able to use such data. The work presented shows it is technically possible. The second step is to show that it allows to get reasonable emissions and then reasonable concentrations. The manuscript provides some elements for this second step.

However what is missing in this work, from my point of view, is the demonstration of the interest of the proposed methodology in comparison to previously existing methods. It could have been relevant to compare a simulation with the emissions derived from the new methodology to a simulation with emissions derived from time-averaged data and applying hourly, daily and monthly factors as often applied within Top-Down approaches. Similarly the comparison to spatially-averaged data (at a chosen grid cell scale and per type of ways) would be of interest.

Reply:

Thank you for your helpful suggestion. We had known that this is important to compare our results with other existing emission inventories as many details as we can, no matter those inventories were establish by top-down or bottom-up method. However, due to the lack of data of the other inventories, some comparisons are unable to achieve in this study. Even so, we had tried out best to add more details about the comparison between our results and others. In section 4.1.2, we had compared the spatial distribution with other two inventories. In page 7, line 26-32, "*Moreover, the spatial distributions of these three emission inventories were compared in this study. Figure 10 shows the distribution of CO from the three different inventories. In urban areas, the results of both MEIC-2016 and PRD-2015 showed the urban areas as emission hotspots. However, the results from the ROE model were much lower for such areas. This may be due to the fact that the ROE model considers the traffic control policies, while the other two inventories do not. In suburban town centers, especially in the eastern and southern parts of Guangzhou, all three inventories showed the same results, namely that these areas were large contributors of on-road emissions. Notably, highways and arterial roads also contributed high emissions in all three inventories.*"

"*However, it should be noted that this comparison was only preliminary; the spatial resolutions of three inventories are inconsistent. Moreover, due to the lack of temporal information in the other two emission inventories, a comparison of the temporals difference could not be conducted. Future studies should focus on improving the accuracy of such comparisons.*" (In page 9, line 37-40)

[Figure]

[Figure]

[Figure]

**Figure 1. Spatial distribution of CO from (a) ROE model, (b) MEIC-2016 and (c) PRD-2015 in Guangzhou.**

2. p3 line 4: Strictly speaking it is not the case for all air pollutants. This sentence could be rewritten avoiding this useless generalisation.

Reply:
Thanks for pointing out this. We had revised the sentence by "*Many studies have successfully applied the regional-scale CTMs to investigate the impact of on-road vehicles on the air quality in urban areas in the regional scale (~100 km).*" in page 3, line 4-5.

3. p3 line 26: This sentence should be rewritten to clarify what is available currently (it is always possible to develop a model to extend its functionalities).

Reply:
Thanks for your constructive comment. We had revised by adding a sentence "*The current version of the ROE model includes the crawler module for the from amap.com (also called the Gaode map) application (Figure 2), a widely adopted map application in China (additional details are provided in section 2.4.).*" to clarify the available data source in the current version of model in page 3, line 38-40.

4. p4 line 15 and section 2.2: More details on the emission factors building methodologies would be useful to appreciate their relevance in a near real-time / "instantaneous" framework. Does their temporal representativeness is fully consistent with the fine temporal description of the traffic data? If not, what are the expected impact on the results?

Reply:
Thanks for pointing out this. We had shown the emission factors and their correction factors in supplementary materials section S1. We had considered the traffic condition correction factors (Table S9 and Table S10) when we calculated the on-road emissions. The emission factors are different under different traffic conditions and the classification of these traffic conditions are based on the real-time traffic data. These correction factors were clarified in page 4, line 22-25, "*The*

*correction factors involving environmental conditions (e.g., temperature, relative humidity, and altitude) and traffic conditions obtained from the technical guide were considered in the study. They are listed in Tables S4–S10 in the supplementary materials. These correction factors were applied to reduce the effects of uncertainties associated with the emission factors.*"

**Table S9. Traffic condition correction factors of petrol vehicle**

| | Traffic Speed (km/h) | | | | |
|---|---|---|---|---|---|
| | <20 | 20–30 | 30–40 | 40–80 | >80 |
| CO | 1.69 | 1.26 | 0.79 | 0.39 | 0.62 |
| HC | 1.68 | 1.25 | 0.78 | 0.32 | 0.59 |
| $NO_x$ | 1.38 | 1.13 | 0.90 | 0.96 | 0.96 |
| $PM_{2.5}$, $PM_{10}$ | 1.68 | 1.25 | 0.78 | 0.32 | 0.59 |

**Table S10. Traffic condition correction factors of diesel vehicle**

| | Emission Standard | Traffic Speed (km/h) | | | | |
|---|---|---|---|---|---|---|
| | | <20 | 20–30 | 30–40 | 40–80 | >80 |
| CO | Pre-China I to China III | 1.43 | 1.14 | 0.89 | 0.84 | 0.61 |
| | China IV, China V | 1.29 | 1.10 | 0.93 | 0.70 | 0.61 |
| HC | Pre-China I to China III | 1.41 | 1.13 | 0.90 | 0.61 | 0.41 |
| | China IV, China V | 1.38 | 1.12 | 0.91 | 0.64 | 0.48 |
| $NO_x$ | Pre-China I to China III | 1.31 | 1.08 | 0.93 | 0.74 | 0.66 |
| | China IV, China V | 1.39 | 1.12 | 0.91 | 0.60 | 0.28 |
| $PM_{2.5}$, $PM_{10}$ | Pre-China I to China III | 1.22 | 1.08 | 0.93 | 0.71 | 0.49 |
| | China IV, China V | 1.36 | 1.12 | 0.91 | 0.65 | 0.48 |

5. p5 line 33: The table 2 only shows the global results without any analysis. I guess a comprehensive comparison of the three inventories is beyond the scope of the current paper, however some general considerations and analysis concerning the discrepancies between the three database appears mandatory for this manuscript.

Reply:
Thank you for your helpful suggestion. We had added some details about the MEIC and PRD inventories. In page 6, line 29-34, "*These two emission inventories used the top-down method to establish on-road emission inventories. Unlike the bottom-up method used in this study, these two inventories first calculated the total emissions based on the VKT data of vehicle categories. In the MEIC inventory, the total number of vehicles was obtained from the relationship between total vehicle ownership and economic development (Zheng et al., 2014), while the PRD inventory acquired information on the number of vehicles from the city-level statistics Yearbook. Then, the spatial distribution of these two inventories was established based on the road network density.*"

According to the uncertainty analysis of emission factors, the uncertainty of $PM_{2.5}$ and $PM_{10}$ is much smaller than the gaseous emissions, leading the large difference of gaseous emissions.

As for NO$_x$ emissions, we thought that the higher NO$_x$ estimate could be due to our updated LPG bus emission factor based on the local study (Zhang et al., 2013). The NO$_x$ emission factor of an LPG-fueled bus is 1.7 times that of a diesel-fueled bus. This maybe one of the reasons leading the higher NO$_x$ estimate. From figure 9, the results showed that the NO$_x$ emission distribution of bus in urban and suburban area was 20.5% and 10.8%.

We had added this content in page 6, line 38 to page 7, line 4, *"the difference of PM$_{2.5}$ and PM$_{10}$ amount was smaller than other gaseous emissions among different inventories. This was because that the uncertainty of particulate matter emission factors was lower than the corresponding values of the other emissions, which led to the large difference for the gaseous emissions and the smaller differences for PM$_{2.5}$ and PM$_{10}$. For NO$_x$ emissions, however, this study showed a higher NO$_x$ estimate than that in the other two inventories. One of the reasons for the higher NO$_x$ estimate may be the application of the updated LPG bus emission factors in this study. Based on a previous local emission factor study, the NO$_x$ emission factor of an LPG-fueled bus is 1.7 times that of a diesel-fueled bus in Guangzhou (Zhang et al., 2013). The results in Figure 8 show that the NO$_x$ emissions distribution attributable to buses in urban and suburban areas were 20.5% and 10.8% of the total NO$_x$, respectively, showing that the LPG-fueled buses may be responsible for higher NO$_x$ estimates in this study compared to those in the other two inventories."*

6.  p5 line 35 and followings: The numbers provided in tables should not be recalled in the body text.

Reply:
Agreed. We had deleted it in section 4.1.1.

7.  p7 line 14-15: From section 3 I understand the "boundary conditions" are considered to feed the MUNICH runs. It implies that other sources than on-road emissions are implicitly considered.

Reply:
Yes. The "boundary conditions" represented the "background concentrations" from outside the simulation street network. To make it understand more easily, we had changed the "boundary conditions" to "background concentrations" in page 6, line and line 21.

8.  p8 line 35: One of the traditional aim of models is to be used for prospective (long term forecast) studies. Could the authors provide some hints on how their methodology could be extended too perform such study?

Reply:
This is a good point. In discussions and conclusions part, we had discussed the possibility of applying the street-level air quality model in forecasting the variation of pollutants. In page 10, line 24-28, *"Recently studies had shown that traffic forecasting models are effective within cities (Min et al., 2009; Cortez et al., 2012; Vlahogianni et al., 2014). These models allow one to obtain predicted traffic-based on-road emissions. Combined with the meteorological forecasting systems and regional air quality forecasting systems, which provide the meteorological and background*

*concentration predictions, respectively, street-level air quality models could be used for street-level air quality forecasting as well.".*

**Response to editor**

**Editor**

Thank you for your helpful advices about the code we uploaded. Our code now is available on GitHub (https://github.com/vnuni23/ROE) and Zenodo (https://doi.org/10.5281/zenodo.3264859). The users who interested in our ROE model could easily download the code from these two websites.

Additionally, it is our mistake that we did not upload any guide or documentation which could guide the users to use our model. We now have uploaded the user guide and some template files to the model package to make it easier for users to understand and use our model. Some plotting scripts are placed in the code repository as well. User guide for the ROE model are also shown in the supplement to the AC3 comment (https://www.geosci-model-dev-discuss.net/gmd-2019-74/gmd-2019-74-AC3-supplement.pdf).

**Reference**

[revised manuscript text omitted]
 | 2.9 | 6.3 | -2.6 | -4.9 | -15.0 | 0.5 | -4.0 | -1.6 | -5.7 | -10.6 | -0.4 | 12.4 | -15.3 | -0.1 | 3.7 |

[Figure]

**Figure 1. The structure of ROE model.**

[Figure]

5    **Figure 2. Traffic information from Gaode map.**

[Figure]

**Figure 3. Traffic control area.**

[Figure]

5      **Figure 4. Diurnal variation of average traffic speed and traffic volume in (a, b) urban area and (c, d) suburban area during weekday and weekend.**

[Figure]

**Figure 5. The percentage of (a) vehicle classification and (b) emission standard.**

[Figure]

**Figure 6. Fuel type percentage of each vehicle classification.**

[Figure]

**Figure 7. Simulation domain from regional scale to street-level scale: (a) four-nested simulation for WRF; (b) domains 3 and 4 covering the Pearl River Delta Region and Guangzhou city, the innermost box corresponds to the Tianhe District; (c) 31 street segments and two observation sites (triangle) within the MUNICH study domain.**

[Figure]

**Figure 8. Emission contribution of each vehicle classification in (a) urban area and (b) suburban area.**

[Figure]

**Figure 9. Spatial distribution of (a) CO, (b) NO$_x$, (c) HC, (d) PM$_{2.5}$, and (e) PM$_{10}$ from the on-road emissions in Guangzhou (blue lines: highways; green lines: arterial roads; local roads are not shown).**

[Figure]

**Figure 10. Spatial distribution of CO from (a) ROE model, (b) MEIC-2016 and (c) PRD-2015 in Guangzhou.**

[Figure]

**Figure 11. The spatial distribution of weekday (a) NO$_x$ and (b) HC emission in the simulated street network.**

[Figure]

**Figure 12. Time series of (a) NO$_2$ and (b) O$_3$ during the simulation period. (black solid line: observation; red dashed line: simulation).**

[Figure]

**Figure 13. The diurnal variation of the total traffic volume in the simulation street network (solid line: normal weekday; dashed line: national holiday; dotted line: normal weekend).**

[Figure]

**Figure 14. The (a) NO$_x$ and (b) O$_3$ diurnal variation of different sensitivity cases in the simulation street network**